# Adaptive Estimation and Optimal Control in Offline Contextual MDPs

**Riddhiman Bhattacharyya**                                     *rbhatta6@ucsc.edu*
*Department of Statistics,*
*University of California, Santa Cruz*
*Santa Cruz, California, USA*

**Sayak Chakrabarty**                         *sayakchakrabarty2025@u.northwestern.edu*
*Department of Computer Science*
*Northwestern University*
*Evanston, IL, USA*

**Imon Banerjee**                                     *imon.banerjee@northwestern.edu*
*Department of Industrial Engineering and Management Sciences*
*Northwestern University*
*Evanston, IL, USA*

**Reviewed on OpenReview:** *https://openreview.net/forum?id=FGBZ4q1HPZ*

## Abstract

Contextual MDPs are powerful tools with wide applicability in areas from biostatistics to machine learning. However, specializing them to offline datasets has been challenging due to a lack of robust, theoretically backed methods. Our work tackles this problem by introducing a new approach towards adaptive estimation and cost optimization of contextual MDPs. This estimator, to the best of our knowledge, is the first of its kind, and is endowed with strong optimality guarantees. We achieve this by overcoming the key technical challenges evolving from the endogenous properties of contextual MDPs; such as non-stationarity, or model irregularity. Our guarantees are established under complete generality by utilizing the relatively recent and powerful statistical technique of $T$-estimation (Baraud, 2011). We first provide a procedure for selecting an estimator given a sample from a contextual MDP and use it to derive oracle risk bounds under two distinct, but nevertheless meaningful, loss functions. We then consider the problem of determining the optimal control with the aid of the aforementioned density estimate and provide finite sample guarantees for the cost function.

## 1 Introduction

Contextual Markov decision processes (contextual MDPs; Hallak et al., 2015) are a core abstraction for sequential decision making with exogenous information ("context") that modulates the dynamics and costs. They underpin applications across healthcare, economics, and operations where learning a decision rule from historical logs—*offline reinforcement learning (offline RL)*—is often the only viable option. Yet, despite their practical importance, a general, assumption-light statistical theory for *estimating the transition mechanism and optimizing costs* in contextual MDPs has remained elusive: most existing approaches rely on stationarity/ergodicity, parametric modeling, or strong smoothness assumptions that are frequently violated in real data. Motivated by the recent success of model based offline RL (Agarwal et al., 2020; Li et al., 2022a) in the finite state-control setting, we aim to achieve the following objectives in this paper.

**Goal.** We develop a non-parametric framework for *offline* contextual MDPs that (i) adaptively estimates the transition density without stationarity or ergodicity assumptions, (ii) delivers *oracle risk bounds* in an empirical Hellinger metric, and (iii) transfers these guarantees to *cost minimization* and *optimal control*

*selection* via a plug-in scheme. Our results close a gap between the realities encountered in practice and existing theory by providing finite-sample guarantees under minimal conditions while remaining minimax-optimal (up to logarithmic factors) under standard smoothness regimes.

**Our Approach.** Let $s(x, a, g, y)$ denote the (contextual) transition density. We construct an estimator $\hat{s}$ by minimizing a penalized *pairwise comparison* functional over a rich, countable model class $\mathcal{S}$:

$$\hat{s} \in \operatorname*{argmin}_{f \in \mathcal{S}} \, \sup_{f' \in \mathcal{S}} \left\{ \alpha \, \mathcal{H}^2(f, f') + T(f, f') \right\} + \frac{L \, \Delta_{\mathcal{S}}(f)}{n},$$

where $\mathcal{H}^2$ is the empirical Hellinger loss, $T(\cdot, \cdot)$ compares candidates on the observed trajectory (including a density-correcting term), and $\Delta_{\mathcal{S}}$ is a data-independent complexity penalty. This is the $T$-estimation principle: first introduced in a seminal PTRF paper by Baraud & Birgé (2009), and rooted in modern model-selection theory, yields robust performance under dependent, non-stationary data and accommodates non-parametric model classes (e.g., piecewise-constant partitions, spline bases etc.).

**Intuition.** Technically, our analysis relies on (i) a martingale Bernstein inequality to control empirical deviations *without* stationarity or mixing assumptions, and (ii) the polarization identity to relate risk in squared-root density space to Hellinger loss. These tools allow us to derive high-probability comparisons that convert into *expected* oracle inequalities after integrating the deviation parameter.

**From density estimation to control:** Given a user-specified, positive cost function $L(x, a, g, y)$, we study the *empirical cost*

$$\hat{C}_n(a, g) \; = \; \int L(x, a, g, y) \, \hat{s}(x, a, g, y) \, \lambda_n^{(1)}(dx, dy),$$

and show that optimizing a simple upper-confidence surrogate of $\hat{C}_n$ identifies (nearly) optimal controls from logged data. Crucially, our guarantees are *policy-agnostic*: any procedure that optimizes the plug-in cost inherits our rates, thereby decoupling statistical estimation from control optimization.

**Remark 1.** *Let $L(x, l, g, y) = |\varphi(g, l)^T(\psi(x) - \psi(y))|$ (Belogolovsky et al., 2021; Zhou et al., 2024), where $\varphi$ and $\psi$ are feature maps and suppose that $\tilde{\omega}_\psi$ is a joint measure between $\mu_\chi(\psi^{-1})$ with itself. Then the cost function defined as above is essentially a cost for a transport from $(\chi, \mu_\chi)$ to itself. In fact, it is exactly the Wasserstein-1 distance if $\tilde{\omega}_\psi$ is the optimal joint measure. Then the problem translates into selecting a control $l$ for a given context $g$ such that "some" risk metric is minimised (see Section 4.3 for more details).*

We now detail the two distinct challenges facing us:

1. Our first challenge is innate to non-parametric inference, and can be linked to "bandwidth selection" for kernel density estimators. For any $\sigma \in (0, 1]$, let $\mathbb{H}^\sigma(\chi)$ be the class of all Hölder smooth functions on $\chi$ (formally defined in Definition 2. See also Bergh & Löfström (1976)). Then, it is known that if $s(x, l, g, \cdot) \in \mathbb{H}^{\sigma_{x,l,g}}(\chi)$, setting the bandwidth $\mathcal{O}(n^{-1/(2\sigma_{x,l,g}+1)})$ one can recover the minimax rate of estimating the density $s(x, l, g, \cdot)$. However, such assumptions are impractical when there are uncountably infinite many $x, l, g$'s.

   At this point, we remark that one cannot simply take the bandwidth to be $\mathcal{O}(n^{-1/(2\sigma+1)})$ where $\sigma = \sup_{x,l,g} \sigma_{x,l,g}$ since, without additional assumptions on the diameter of $\chi$, Hölder spaces generally **do not** satisfy the embedding $\mathbb{H}^\alpha(\chi) \subset \mathbb{H}^\beta(\chi)$ for $\alpha < \beta$. Therefore, this problem of bandwidth selection is even more tenuous in the MDP context.

2. Our second challenge is on the collection of historical data. For regular (non-contextual) MDPs it is common practice to assume that the actions $a_i$ depend only on the current state $X_i$ (Sutton & Barto, 2018), and the contexts $G_i$ are independent; an assumption which is easily violated. As an example, note that, hidden Markov models (which may correspond in real-life to the biological markers for the evolving health conditions of a single patient), are not Markovian up to any degree. However, since the contexts $G_i$'s can themselves arise out of such an evolving time series, it is desirable to have an estimation procedure that is "robust" to non-stationarity or non-ergodicity but optimal if such extra conditions are available. We now informally state our contributions.

**Contributions.**

- **Non-parametric estimation for contextual MDPs.** We introduce a $T$-estimator for the transition density that requires only boundedness of a dominating density and *no* stationarity or ergodicity. We prove an *oracle inequality* in empirical Hellinger loss:

$$\mathrm{E}\big[\mathcal{H}^2(s, \hat{s})\big] \;\lesssim\; \inf_{f \in \mathcal{S}} \left\{ \mathcal{H}^2(s, f) + \tfrac{\Delta_{\mathcal{S}}(f) + \dim(f) \log n}{n} \right\}.$$

- **Minimax-rate adaptivity.** For standard smoothness classes (e.g., Hölder/Besov), our procedure attains the optimal non-parametric rate (up to poly-log order): $\mathrm{E}[\mathcal{H}^2(s, \hat{s})] \lesssim \big(\tfrac{\log n}{n}\big)^{\bar{\sigma}/(4+\bar{\sigma})}$.

- **Cost minimization with finite-sample guarantees.** We establish non-asymptotic bounds for the *plug-in* cost gap between the control selected by our estimated surrogate for the cost function and the true optimal control. The bound scales as $O\Big(\sqrt{\tfrac{\max(\Delta_{\mathcal{S}}(f), \log n)}{n}}\Big)$.

*Note for Practitioners:* Our results provide a practical, assumption-light route to *offline* cost-sensitive decision making in contextual MDPs: learn a non-parametric transition model with $T$-estimation, plug it into the task-specific cost, and optimize controls with the estimated costs. This pipeline yields rigorous risk and optimality guarantees without relying on stationarity, ergodicity, or parametric misspecification—a favorable regime for real-world logs with shifting contexts and irregular dynamics.

*The rest of the paper is organised as follows:* Section 2 outlines a comprehensive discussion of relevant research works. In section 3, we formally introduce the model selection procedure and the relevant notations. In section 4, we provide our key theoretical results on estimating $s$, with the proof sketches for these, while the full proofs have been deferred till the appendix due to lack of space. Section 5 contains our theoretical guarantees on estimating the optimal cost, and finally, in Section 8, we conclude by mentioning the limitations and broader impact of our work. All technical proofs are deferred to the Appendix.

## 2 Background and Related Research

Initial works on model selection as a form of estimation dates back to Barron et al. (1999), and for a general overview of the literature on model selection, we refer the readers to the exceptional (albeit slightly dated) monograph by Pascal Massart (Massart, 2007). $T$-estimators, specifically, have seen a significant amount of focus in recent years. They have been used as a general density estimation procedure in i.i.d. (Baraud & Birgé, 2009; Birgé, 2006), bivariate (Sart, 2017), Markovian (Sart, 2023) and controlled Markovian (Banerjee et al., 2025b) contexts, and seen applications in machine learning related fields like differential privacy (Sart, 2023).

In the interest of exposition, we point out that recently, $T$-estimators have been generalised in a series of groundbreaking papers starting with Baraud et al. (2017), with follow ups in Baraud & Birgé (2018), and Baraud & Birgé (2020) These so-called $\rho$-estimators—based on the Bhattacharya correlation coefficient $\rho(f, g) := \int \sqrt{fg} \, d\lambda_n$—are non-parametric, yet MLE-like in their efficiency. However, theory of $\rho$-estimation is still in its infancy, and for the sake of brevite, we fallback on the various technical tools available to us for $T$-estimators.

Contextual MDPs—first formally introduced in Hallak et al. (2015)—have become a cornerstone of cost optimisation and decision making in diverse fields such as finance, economics, and healthcare (Batsis & Samothrakis, 2024; Xiao et al., 2019; Tang & Wiens, 2021; Li et al., 2022b; Javanmard & Nazerzadeh, 2019), in both online (Li et al., 2022b) and offline (Zhou et al., 2024) settings; proving to be a major tool in solving problems with real life applications (Cao et al., 2023). Much work has been done in the field of contextual MDPs with PAC/sample complexity bounds for the optimal policy (Krishnamurthy et al., 2016; Sun et al., 2019; Jiang et al., 2017). However, our work is more general; as we prove in Theorem 2, one can use any method to find the optimal control and leverage rate optimality guarantees (as given Theorem 3) endowed by our estimator—a crucial setting which till now had remained largely unexplored, formalized by the following open question.

**Open question.** The paper resolves the following open questions in the theory of contextual MDPs: (i) Can we provide a *complete convergence theory for contextual MDPs*—oracle risk bounds for transition function and the optimal cost without prior assumptions on the data generating process? (ii) Can we then, under further suitable assumptions, derive minimax optimality guarantees for our estimator and the associated cost function?

With that, we move on to formally define our problem.

## 3 Problem Formulation

We initiate this section by introducing some notation that will be used repeatedly throughout the paper. Let $\mathbb{N}$ and $\mathbb{R}$ denote the natural and real numbers, and the symbol $\lfloor \cdot \rfloor$, the floor function. All random variables in this paper will be defined with respect to a filtered probability space $(\Omega, \mathcal{F}, \mathbb{F}, \mathbb{P})$, where $\mathcal{F}$ is a $\sigma$-algebra and $\mathbb{F} := \{\mathcal{F}_i\}_{i \geq 0}$, with $\mathcal{F}_i \subset \mathcal{F}$, is a given filtration. Let $\{(X_i, a_i, G_i)\}$ represent a discrete-time stochastic processes adapted to $\mathbb{F}$, and taking values in $\chi \times \mathbb{I} \times \mathcal{G} \subseteq \mathbb{R}^{d_1} \times \mathbb{R}^{d_2} \times \mathbb{R}^{d_2}$. We call $\chi$, $\mathbb{I}$, and $\mathcal{G}$ the *state, control,* and *context* spaces respectively. For all non-negative integers $i, j$, we define $\mathcal{H}_i^j := (X_j, a_j, G_j \ldots, X_i, a_i, G_i)$ and $\hbar_i^j := (x_j, l_j, g_j, \ldots, x_i, l_i, g_i)$ and note that $\hbar_i^j$ is an element of $(\chi \times \mathbb{I} \times \mathcal{G})^{j-i+1}$. The $\sigma$-field generated by $\mathcal{H}_i^j$ shall be $\mathcal{F}_i^j$. Throughout the paper, we will assume that $\chi$, $\mathbb{I}$, and $\mathcal{G}$ are compact. $\mathbb{X} := \chi \times \mathbb{I} \times \mathcal{G} \times \chi$ shall denote the augmented state-space, and is compact by the previous assumption. When $\mathbb{X}$ is *not compact*, all of our theory still continues to hold on any restriction of $\jmath$ on a compact subset $A \subset \chi \times \mathbb{I} \times \chi$, given by $\jmath \mathbb{1}_A$. Observe that $\jmath \mathbb{1}_A$ is not necessarily a conditional density, since it may not integrate upto 1.

Let $\mathbb{E}[X]$ be the expectation and $\sigma(X)$ the $\sigma$-algebra induced by $X$. We endow $\chi$, $\mathbb{I}$, and $\mathcal{G}$ with integrating measures $\mu_\chi$, $\mu_\mathbb{I}$, and $\mu_\mathcal{G}$ respectively. One can assume $\mu$'s to be Lebesgue when $\chi, \mathbb{I}, \mathcal{G}$ are Euclidean or the counting measure when they are discrete. By $\text{Vol}(\mathcal{S})$ we denote the volume of the set $\mathcal{S}$ with respect to its natural measure. As an example, if $\mathcal{S} \subset \chi$, then $\text{Vol}(\mathcal{S}) = \mu_\chi(\mathcal{S})$; if $\mathcal{S} \subset \mathbb{I}$, then $\text{Vol}(\mathcal{S}) = \mu_\mathbb{I}(\mathcal{S})$, etc. $\mathcal{C}$ and $c$ are always used to denote universal constants whose values can change from line to line. We call $m = \{k : k \subseteq \chi \times \mathbb{I} \times \chi\}$ to be a *partition* of $\chi \times \mathbb{I} \times \chi$ if $\bigcup_{k \in m} k = \chi \times \mathbb{I} \times \chi$ and $k \bigcap k' = \emptyset$ for all distinct $k, k' \in m$. Finally, to avoid trivialities, we assume throughout the paper that the number of samples, denoted by $n$ is at least 3.

### 3.1 Definitions

Our objective in the paper is to select the best density (the eponymous "model") from a given class of models. To set the stage, we introduce the following definitions.

Let $\lambda_n := \sum_{i=0}^{n-1} \delta_{X_i, a_i, G_i} \otimes \mu_\chi / n$ denote the product between the point measure $n^{-1} \sum_{i=0}^{n-1} \delta_{X_i, a_i, G_i}$ and $\mu_\chi$. Formally, for any set $\mathcal{X} \subset \mathbb{X}$ such that $\mathcal{X} = \mathcal{X}_1 \times \mathcal{X}_2 \times \mathcal{X}_3 \times \mathcal{X}_4$,

$$\lambda_n(\mathcal{X}) = \frac{1}{n} \sum_{i=0}^{n-1} \mathbb{1}[(X_i, a_i, G_i) \in \mathcal{X}_1 \times \mathcal{X}_2 \times \mathcal{X}_3] \mu_\chi(\mathcal{X}_4).$$

Observe that this is a valid measure on $\mathbb{X}$. By $\mathbb{L}_1^+(\mathbb{X}, \lambda_n)$, we denote all positive, absolutely integrable functions on $\mathbb{X}$. For $f_1, f_2 \in L_1^+(\mathbb{X}, \lambda_n)$ (not necessarily densities) on $\mathbb{X}$, we define the square of the **empirical Hellinger distance** $\mathcal{H}^2$ as

$$\mathcal{H}^2(f_1, f_2) := \frac{1}{2} \int_\chi \left( \sqrt{f_1} - \sqrt{f_2} \right)^2 d\lambda_n.$$

It follows that $\mathcal{H}$ is a nonnegative random variable adapted to $\mathcal{F}_0^n$. Our model selection procedure is as follows: For some universal constant $\alpha$, let $L > 0$, and $\mathcal{S} \subset \mathbb{L}_+^2(\mathbb{X}, \lambda_n)$ be a countable (but possibly infinite) class of functions. The map $\Delta_\mathcal{S} : \mathcal{S} \to \mathbb{R}$ is said to be a penalty on $\mathcal{S}$ if $\Delta_\mathcal{S}(f) > 0 \ \forall f \in \mathcal{S}$. For convenience of notation, we define

$$\varphi(x, y) := \frac{\sqrt{y} - \sqrt{x}}{2\sqrt{x + y}}$$

For any two functions $f_1, f_2 : \chi \times \mathbb{I} \times \mathcal{G} \times \chi \to \mathbb{R}$ define $T(f_1, f_2)$ as,

$$T(f_1, f_2) := \frac{1}{n} \sum_{i=0}^{n-1} \underbrace{\int \frac{1}{\sqrt{2}} \varphi(f_1, f_2) \delta_{X_i, a_i, G_i, X_{i+1}}}_{A}$$

$$+ \underbrace{\int \sqrt{\frac{f_1 + f_2}{2}} \cdot (\sqrt{f_2} - \sqrt{f_1}) \, d\lambda_n}_{B} + \underbrace{\int (f_1 - f_2) \, d\lambda_n}_{C}. \tag{3.1}$$

**Remark 2.** *T can intuitively be thought of as a comparison of $f_1$ and $f_2$, with A comparing which one fits the $\{(X_i, a_i, G_i, X_{i+1})\}$ process better, B comparing which one fits the $\{(X_i, a_i, G_i)\}$ process better, and C penalising if $f_1$ or $f_2$ is not a proper density.*

**Definition 1.** *$\vartheta$ is said to be a model-selection procedure on $\mathcal{S}$ if*

$$\vartheta(f) = \sup_{f' \in \mathcal{S}} \left[ \alpha \mathcal{H}^2(f, f') + T(f, f') - L \frac{\Delta_{\mathcal{S}}(f')}{n} \right] + L \frac{\Delta_{\mathcal{S}}(f)}{n}. \tag{3.2}$$

Our choice of estimator shall be the following:

$$\hat{\jmath} := \operatorname*{argmin}_{f \in \mathcal{S}} \vartheta(f) + \frac{1}{n}. \tag{3.3}$$

Observe that $\hat{\jmath}$ is precisely the minimum-contrast estimator of Massart (2007) and, following the foundational PTRF paper Barron et al. (1999), is estimated via a penalised selection procedure from a model class. Recently, this estimator has been used with great success in model selection for Markov chains (Sart, 2014), and in the following section we investigate its properties in the contextual MDP setting.

**Remark 3.** *$\hat{\jmath}$ depends on n and L and may not be unique. In that case, all of the choices are valid estimators.*

**On the choice of the model class $\mathcal{S}$.** In practice, the performance of the estimator depends on the complexity of the candidate model class $\mathcal{S}_\ell$. The precise choice is somewhat ambiguous but Larger model classes typically yield more precise estimation, with the obvious tradeoff of under-penalizing bad models for small sample sizes.

- For smaller sample sizes, smaller classes may be beneficial to avoid excessive bias with the added risk of under-penalizing if the model class is too large.

- For moderate sample sizes, setting standard model classes like those in examples below (see Example 1 or 2) work well.

- For larger sample sizes, even richer classes may suffice.

## 4 Theoretical Results

Before stating our main theorem we state the following assumptions (and discuss them below)

**Assumption 1.** *Any $f \in \mathcal{S}$ is finite dimensional (over the $\mathbb{L}^2$ norm) with dimension $\dim(f)$. We further assume that $\sup_{f \in \mathcal{S}} 0 \leq f \leq 1$ and $\sum_{f \in \mathcal{S}} e^{-\Delta_{\mathcal{S}}(f)} \leq 1$.*

**Assumption 2.** *For all $i \in \{0, \ldots, n-1\}$, $(X_i, a_i, G_i, X_{i+1})$ admits a (possibly inhomogenous) density $\phi_i$ with respect to some known measure $\mu_{\mathbb{X}}$ (defined formally in Section A.5) such that $\phi(\cdot, \cdot, \cdot, \cdot) \leq \kappa$ for some constant $\kappa > 0$ and $\mu_{\mathbb{X}}(\mathbb{X}) = 1$.*

The first assumption signifies that the penalty must be large enough such that $e^{-\Delta_{\mathcal{S}}(f)}$ is summable on $f$; the upper bound of 1 is without losing generality. We discuss this further with an example below.

The second assumption has two parts. The requirement on the density is mild since it assumes an *upper bound* on the density, as opposed to the more prevalent (and somewhat tenuous) *lower bound* that is ubiquitous adaptive learning literature (see for instance A5, Lacour (2007) or Assumption 4.1 in Sart (2014).) We can now state our main oracle risk bound.

### 4.1 Oracle Risk Bound for Empirical Hellinger

**Theorem 1.** *Under the conditions of Assumption 1, there exists an universal constant $L_0 > 0$ such that if $L \geq L_0$, the estimator $\hat{s}$ satisfies*

$$\mathcal{C}\mathbb{E}\big[\mathcal{H}^2(s, \hat{s})\big] \leq \mathbb{E}\left[\inf_{f \in \mathcal{S}} \left\{\mathcal{H}^2(s, f) + L\frac{\Delta_{\mathcal{S}}(f)}{n}\right\}\right].$$

Theorem 1 is standard in $T$ estimation literature, some key references for which are (Baraud, 2011; Sart, 2014; Baraud et al., 2017). Classical approaches such as regularised MLE, kernel density estimation (KDE), or fitted Q-iteration typically rely on strong structural assumptions, e.g. stationarity, ergodicity, or smoothness of the transition kernel. As mentioned in Section 1, these assumptions are often violated in contextual MDPs with irregular or non-stationary dynamics, rendering such methods either inconsistent or suboptimal. To the best of our knowledge, $T$-estimators are the only estimators capable of accommodating both non-stationarity and irregularity, and are therefore useful despite their shortcomings and we refer the readers to various literature in adaptive estimation (Massart, 2007; Baraud & Birgé, 2009; Baraud, 2011; Sart, 2014) for more details.

Before providing a sketch of the proof, we discuss some implications of the previous theorem. Observe that the statement of the previous theorem makes no assumption on the data generating process, beyond the fact that $X_i$ is Markovian on $(X_{i-1}, a_{i-1}, G_{i-1})$. Theorem 1 can then be interpreted as follows: $\hat{s}$ is the best estimator one can obtain for *a given sample* $(X_{i-1}, a_{i-1}, G_{i-1})$, on a *given model class $\mathcal{S}$*. There are various interesting choices for $\mathcal{S}$, and some are given in Section 4.2. This results in us providing settings where our estimation method works and gives finite sample guarantees.

*Proof sketch for Theorem 1.* **Step I.** We analyze the sign of

$$T(f, \hat{s}) + L\frac{\Delta_{\mathcal{S}}(f)}{n} - L\frac{\Delta_{\mathcal{S}}(\hat{s})}{n}.$$

If $T(f, \hat{s}) + L\frac{\Delta_{\mathcal{S}}(f)}{n} - L\frac{\Delta_{\mathcal{S}}(\hat{s})}{n} \geq 0$, a direct comparison yields for some universal constants $\alpha$ and $\varepsilon$

$$\alpha\mathcal{H}^2(s, \hat{s}) \ \leq \ (1+\varepsilon)\mathcal{H}^2(s, f) + \tfrac{2L}{n}\Delta_{\mathcal{S}}(f) + \kappa\xi.$$

**Step II.** In the complementary case, using a martingale Bernstein inequality, we derive a lemma on the concentration of the risk metric (Lemma 4) which ensures that, with probability at least $1 - e^{-n\xi}$,

$$(1-\varepsilon)\mathcal{H}^2(s, f') + T(f, f') - L\frac{\Delta_{\mathcal{S}}(f')}{n} \leq (1+\varepsilon)\mathcal{H}^2(s, f) + L\frac{\Delta_{\mathcal{S}}(f)}{n} + \kappa\xi$$

for all $f' \in \mathcal{S}$. This further implies where $\nu = (1-\varepsilon)/\alpha - 1$

$$\alpha\mathcal{H}^2(f, f') \ \leq \ (2+\varepsilon+\nu^{-1})\mathcal{H}^2(f, s) + \frac{2L}{n}\Delta_{\mathcal{S}}(f) + \kappa\xi + \frac{1}{n}.$$

**Step III.** Combining the bounds from the two cases with the Hellinger triangle inequality

$$\alpha\mathcal{H}^2(s, \hat{s}) \ \leq \ 2\alpha\mathcal{H}^2(s, f) + 2\alpha\mathcal{H}^2(f, \hat{s}),$$

and applying the above control on $\mathcal{H}^2(f, \hat{s})$, we obtain with probability at least $1 - e^{-n\xi}$

$$\mathcal{C}\mathcal{H}^2(s, \hat{s}) \ \leq \ \mathcal{H}^2(s, f) + L\frac{\Delta_{\mathcal{S}}(f)}{n} + \xi,$$

where $\mathcal{C} = \alpha/\min\{2(2+\alpha+\varepsilon+\nu^{-1}), 2\kappa\}$. Taking complementation and integrating both sides by $\xi$ now yields the final result. □

To state the following corollary we need one further definition concerning the squared loss function.

$$d^2(f_1, f_2) := \int_{\mathbb{X}} (\sqrt{f}_1 - \sqrt{f}_2)^2 d\mu_{\mathbb{X}}.$$

The proof of the following corollary can be found in Section A.5

**Corollary 1.** *Let $L \geq L_0$ for some universal constant $L_0$ and assume that the hypothesis in Assumptions 1, and 2 holds. Then, the selected estimator as defined in 3.3 satisfies*

$$\mathcal{C}\mathbb{E}\left[\mathcal{H}^2(\jmath, \hat{\jmath})\right] \leq \inf_{f \in \mathcal{S}} \left\{ d^2(\sqrt{\jmath}, f) + \frac{\Delta_{\mathcal{S}}(f) + \dim(f)\log n}{n} \right\}$$

*for some large enough constant $\mathcal{C}$ depending only on $\kappa$.*

Results in the flavor of Corollary 1 are standard in $T$ estimation literature and we point the readers to various previous literature for more details (Sart, 2014; Baraud et al., 2017; Birgé, 2006; Sart, 2017).

### 4.2 Examples

At this point, it seems reasonable to ground the abstractions of the previous section into some examples. We begin with some model classes which satisfy the Assumption 1, beginning with dyadic cuts (DeVore & Yu, 1990).

**Example 1** (Piecewise Constant Estimators on Dyadic Cuts)**.** *Let $\chi = \mathbb{I} = \mathcal{G} = [0, 1]$. Then, we define the space of all dyadic cuts recursively. Define $\mathcal{M}_0 := \{[0, 1]^4\}$. For any $\ell \in \mathbb{N}$, let $m \in \mathcal{M}_\ell$ and $k \in m$. Thus $k$ is an element of a partition of $[0, 1]^4$, so $k \subseteq \mathbb{R}^4$. Let $k_1, k_2, \ldots, k_{2^4}$ be the $2^4$ sets obtained by equally dividing $k$ along each axis. Let*

$$S(m, k) := m \cup \{k_1, k_2, \ldots, k_{2^4}\} \setminus k.$$

*Then*

$$\mathcal{M}_{\ell+1} := \left\{ \bigcup_{m \in \mathcal{M}_\ell} \bigcup_{k \in m} S(m, k) \right\} \cup \mathcal{M}_\ell.$$

*The class of all piecewise constant estimators on this class of dyadic partitions is defined to be $\mathcal{S} := \bigcup_{m \in \mathcal{M}} \left\{ \sum_{k \in m} a_k \mathbb{1}_k : a_k > 0 \right\}$ with the corresponding penalty to be defined as $\Delta_{\mathcal{S}}(f) = |m| - (\log 3)/2$, where $|m|$ is the number of constant pieces for the piecewise function $f$. Obviously, $\Delta_{\mathcal{S}}(f) \geq 0$. It is now a standard result from Baraud & Birgé (2009) (see Section 3) that $\sum_{f \in \mathcal{S}} e^{-\Delta_{\mathcal{S}}(f)} \leq 1$.*

It is intuitively clear from the previous construction that there is no benefit to considering piecewise constant estimators with more than $n$ distinct bins. Consequently, our optimal estimator will lie in $\mathcal{M}_\ell$ such that $\ell = o(n)$. For a formalisation and proof of this fact, we refer the readers to Proposition 3 in Banerjee et al. (2025b). The computational cost of finding the estimator is $O(e^{\ell d})$, and we refer the readers to Proposition A.1 in Sart (2014) for this fact.

**Example 2** (Splines)**.** *Let $\varphi^{(1)}, \varphi^{(2)}, \ldots,$ be the orthonormal polynomial basis with respect to the $L_2$ norm, and its corresponding inner product. Let constants $c_i^{(\ell)} \in \mathbb{R}$ induce a triangular array such that for each $\ell$, $\sum c_i^{(\ell)} = 1$ and consider*

$$\mathcal{M}_\ell = \left\{ \sum_{i=1}^{\ell} a_i \varphi^{(i)} : a_i \in \cup_i \{c_i^{(l)}\} \right\}$$

*We now $\mathcal{S} = \bigcup_\ell \mathcal{M}_\ell$ and $\Delta_{\mathcal{S}}(f) = \ell + a$ for $a = \log(\frac{e-2}{2})$ whose specifics are given in Section B.2.*

The following proposition highlights that the functions in Example 2 satisfy Assumption 1. Its proof is deferred to Section B.2

**Proposition 1.** *Consider the class of functions given by $\mathcal{S}$ in example 2 with corresponding penalty $\Delta_{\mathcal{S}}(f)$. Then,*

$$\dim f < \infty \quad \text{and} \quad \sum_{f \in \mathcal{S}} e^{-\Delta_{\mathcal{S}}(f)} \leq 1.$$

### 4.3 Optimality of the Risk Bound

A natural subsequent question is now the optimality of the risk bounds derived in the previous section, which is what we dedicate this section towards. To derive the optimality bounds, we first introduce some notation.

**Definition 2.** *We call a function $f : A \to \mathbb{R}$ to belong to the Hölder space $\mathbb{H}_\sigma(A)$ with parameter $\sigma \in (0, 1]$ and finite norm $\|f\|_\sigma > 0$ if $|f(x) - f(y)| \leq \|f\|_\sigma \|x - y\|^\sigma \ \forall x, y \in A$.*

Recall that $\mathbb{H}^1(A)$ is the space of all Lipschitz smooth functions, and that elements of $\mathbb{H}^\sigma(A)$ are constant functions when $\sigma > 1$.

**Definition 3.** *Given a function $f \in L_p(\mathbb{X}), 0 < p \leq \infty$, and any integer $r$, we defne its modulus of smoothness of order $r$ as*

$$\omega_r(f, t)_p := \sup_{0 < |h| \leq t} \|\Delta_h^r(f, \cdot)\|_{L_p(\Omega)}, \quad t > 0,$$

*where $h \in \mathbb{R}^d$ and $|h|$ is it Euclidean norm. Here, $\Delta_h^r$, is the $r$-th difference operator, defined by*

$$\Delta_h^r(f, x) := \sum_{k=0}^r (-1)^{r-k} \binom{r}{k} f(x + kh), \quad x \in \Omega \subset \mathbb{R}^d,$$

*where this difference is set to zero whenever one of the points $x + kh$ is not in the support of $f$. It is easy to see that for any $f \in L_p(\Omega)$, we have $\omega_r(f, t)_p \to 0$, Then, for any $\sigma \in (0, 1)$, Besov space $\mathbb{B}_q^\sigma(L_p(A))$ consists of all $f$ such functions such that*

$$|f|_{\mathbb{B}_q^\sigma(L_p(A))} := \begin{cases} \int_{t>0} t^{q\sigma - 1}(\omega_r(f, t)_p)^q dt & 0 < q < \infty \\ \sup_{t \geq 0} t^{q\sigma - 1}(\omega_r(f, t)_p)^q & q = \infty \end{cases}$$

*is finite. Then, we define $\mathbb{B}^\sigma(L_p(A))$ as*

$$\mathbb{B}^\sigma(L_p(A)) := \begin{cases} \mathbb{B}_p^\sigma(L_p(A)), & p \in (1, 2) \\ \mathbb{B}_\infty^\sigma(L_p(A)), & p \geq 2 \end{cases}$$

*with the attached norm $\|\cdot\|_{\sigma, p}$.*

**Remark 4.** *With $\sigma \in (0, 1)$, we restrict ourselves to isotropic Besov spaces.*

Without losing generality, let $\mathbb{X} = [0, 1]^4$. The following corollary (which follows similarly to Corollary 4.3 in Sart) establishes the optimality of $\hat{s}$.

**Corollary 2.** *Consider the class of models in Example 1, and the Assumptions 1, and 2. Then, if $\sqrt{s} \in \mathbb{H}^\sigma([0, 1]^4)$ (or $\sqrt{s} \in \mathbb{B}^\sigma(L_p([0, 1]^4))$ ), there exists a constant $\mathcal{C}$ depending only on $\kappa, p$, and $\|\sqrt{s}\|_\sigma$, (respectively $\|\sqrt{s}\|_{\sigma, p}$) such that*

$$\mathcal{C}\mathbb{E}[\mathcal{H}^2(s, \hat{s})] \leq \left(\frac{\log n}{n}\right)^{\frac{\bar{\sigma}}{4 + \bar{\sigma}}}$$

*where $\sigma > 4(1/p - 1/2)\mathbb{1}[(1/p - 1/2) > 0]$.*

## 5 Optimal Control Determination

One objective in obtaining the estimates of the transition probabilities is to learn the optimal control for MDPs bearing a cost for state transitions, where the optimality is considered with respect to this cost of transition between states given a control and a context, $L : \mathbb{X} \to \mathbb{R}_+$. Therefore, given a context $G$, we wish to find the solution

$$a^*(G) = \underset{a \in \mathbb{I}}{\operatorname{argmin}} \ C_n(a, G) \tag{5.1}$$

where $C_n(a, G)$ is as defined below in (5.3). Recall that,

$$\lambda_n := \frac{1}{n} \sum_{i=0}^{n-1} \delta_{X_i, a_i, G_i} \otimes \mu_\chi,$$

and let

$$\lambda_n^{(2)} = \int_{\chi \times \chi} \lambda_n = \frac{1}{n} \sum_{i=1}^n \delta_{a_i, G_i} \text{ and } \lambda_n^{(1)} = \lambda_n / \lambda_n^{(2)}. \tag{5.2}$$

Intuitively, $\lambda_n^{(2)}$ is the empirical distribution of the context–action pairs $(a_i, G_i)$ observed in the dataset, while $\lambda_n^{(1)}$ represents the corresponding conditional distribution of state transitions given a particular context–action pair. In other words, $\lambda_n^{(1)}$ captures how states evolve conditional on $(a_i, G_i)$, and $\lambda_n^{(2)}$ records how often each $(a_i, G_i)$ is observed. This decomposition allows us to express the cost functional as an empirical expectation under the observed data distribution.

Note that $a^*(G)$ can be a set of controls and hence a single optimal control might not be best and in such a scenario any control in these sets are equivalent for our purposes.

Next, as in the introduction, $L : \mathbb{X} \to \mathbb{R}_+$ denotes the cost function and $C_n(l, G)$ denotes the cost of control $l$ for context $G$ annealed over $\lambda_n^{(2)}$. Formally,

$$C_n(l, g) = \int L(x, l, g, y) \jmath(x, l, g, y) \lambda_n^{(1)}(dx, dy). \tag{5.3}$$

This is the *empirical* cost of allocating control $l$ for context $G$. We denote the estimated counterpart of $C$ with $\hat{C}$, i.e.

$$\hat{C}_n(l, g) = \int L(x, l, g, y) \hat{\jmath}(x, l, g, y) \lambda_n^{(1)}(dx, dy) \tag{5.4}$$

and define the following optimization problem

$$\hat{a}^*(G) = \underset{a \in \mathbb{I}}{\operatorname{argmin}} \, \hat{C}_n(a, G). \tag{5.5}$$

which is the estimated analogue of the true optimization problem as defined in (5.1). For usual contextual RL MDPs, one can use any algorithm (we point readers to standard texts like Sutton & Barto (2018)) to find $\hat{a}^*$ like UCB, V-iteration, Q-iteration, policy gradient etc. Our objective will be to exhibit that under certain simple assumptions on the cost function, the recovered control provides the optimal cost under the following mild assumption.

**Assumption 3.** *We assume that the true density $\jmath \in \mathcal{S}$, all densities in $\mathcal{S}$ are linear in $a$ and that the cost function $L$ is bounded by a positive constant $\|L\|_\infty$. Furthermore, we assume that $\Delta(s) < \infty$*

We briefly note that the assumption $\jmath \notin \mathcal{S}$ may be violated in practice by extrinsic factors like distribution shift, and we deal with this case separately in Section 6. We now have the following theorem.

**Theorem 2.** *Let Assumptions 1, and 3 hold. Then for any $\hat{a}_n \in \hat{a}^*(G)$ and $a \in a^*(G)$, one has*

$$\mathbb{E}\left[ \int \left( C_n(\hat{a}_n, G) - C_n(a, G) \right) d\lambda_n^{(2)} \right] = O\left( \sqrt{\frac{\log n}{n}} \right)$$

Theorem 2 implies that upon using our scheme, on average, we select the optimal treatment in most realizations, with the probability of the sub-optimal control(s) choice diminishing to 0 with increasing sample size. In the following paragraphs, we outline the key steps in the proof.

*Proof sketch of Theorem 2.* **Step I.** In the first step, we redefine the problem in terms of the value functions, which are considered as negative cost functions. Then we show

$$-C_n(a^*, G) + C_n(\hat{a}^*, G) \le -C_n(a^*, G) + C_n(\hat{a}^*, G) + \hat{C}_n(a^*, G) - \hat{C}_n(\hat{a}^*, G).$$

**Step II.** For the first term we leverage the definition of cost and the Hellinger distance with the Cauchy-Schwarz inequality to establish

$$\mathbb{E}\left[ \int \left( -C_n(a^*, G) + \hat{C}_n(a^*, G) \right) d\lambda_n^{(2)} \right] \le \|L\|_\infty \sqrt{2} \frac{(1 + \dim(s))}{\mathcal{C}} \sqrt{\frac{\log n}{n}}.$$

**Step III.** We establish the same bound for the second term and establish the proof. $\square$

At this point, we note that the performance guarantee on $\hat{a}^*$ in Theorem 2 is derived using a simple plug-in estimator based on the model $\hat{\jmath}$. It is widely known that plug-in approach works well for reinforcement learning tasks (Agarwal et al., 2020; Zhu et al., 2024). We now show that the rate function derived in Theorem 2 is optimal. To that end, we define the minimax risk. Let $\mathcal{M}$ be the set of all contextual MDPs. Then the minimax risk is defined as

$$\mathcal{R}_n := \min_{\hat{a}^* \in \mathbb{I}} \max_{\mathrm{mdp}^* \in \mathcal{M}} \mathbb{E}\left[\int C_n(\hat{a}^*, G) - C_n(a^*, G)\lambda_n^{(2)}\right]$$

Observe that the rate of Theorem 2 matches the known upper bounds for non-stationary MDPs like contextual bandits Lattimore & Szepesvári (2020), which has a known lower bound $\sqrt{n}$. On the other hand, bandits (even in the offline setting) do not involve transition functions, and the lower bound proofs do not directly apply. Theorem 3 improves upon existing literature by showing that the optimal cost cannot be improved over $\sqrt{n}$. This does so by directly linking the estimation problem for the transition kernel with choosing the optimal control and shows that a mistake is made if the transition kernel is not estimated correctly.

**Theorem 3.** *Let Assumptions 3 hold. Then, minimax risk $\mathcal{R}_n$ satisfies*

$$\mathcal{R}_n \geq \Omega\left(\sqrt{\frac{1}{n}}\right)$$

Observe in contrast with Theorem 2 that Assumption 1 is not relevant towards proving the lower bound. It is only relevant towards proving the upper bound since it places a selection procedure on the class of models. Finally, we make note that Theorem 3 shows that Theorem 2 is *rate optimal*, but it is unclear whether it is also optimal on the model parameters. It remains an important open question for future studies in this direction.

**Remark 5.** *We remark that the minimax risk is achieved by a plug-in estimator which was previously known in finite state-control spaces (Agarwal et al., 2020; Li et al., 2022a). Our results therefore, extend this important body of literature by extending it to compact (but possibly infinite) state-control spaces.*

## 6 Offline Policy Evaluation and Distribution Shift

Offline policy evaluation (OPE) is a key problem in offline RL settings. In this section, we show how the results of the previous section can be used for offline policy evaluation and then extend those results to the setting where the data faces distribution shift. Let $\Delta(\mathbb{I})$ denote the probability simplex on the control space, and suppose $\pi : \chi \times \mathcal{G} \to \Delta(\mathbb{I})$ is a given stationary stochastic policy. We only consider discounted MDPs so that a stationary policy stays optimal (Bertsekas, 2011). Let $\lambda_n^{\mathcal{G}} := \frac{1}{n}\sum \delta_{G_i}$, and $\lambda_n^{\chi} := \sum \delta_{X_i}$ . Recall the definition of integral operators (Bakry et al., 2014) and observe that under policy $\pi$, the true transition operator and the plug-in transition operator on the space of bounded continuous functions are given by

$$P_\pi f(x) := \int \int f(y)\,\jmath(x, \pi(x, g), g, y)\, d\lambda_n^{\mathcal{G}} d\mu_\chi,$$

and

$$\hat{P}_\pi f(x) := \int \int f(y)\,\hat{\jmath}(x, \pi(x, g), g, y)\, d\lambda_n^{\mathcal{G}} d\mu_\chi,$$

where $f$ is any bounded continuous function. For simplicity, we assume that $\hat{\jmath}$ is a density, which implies that $P_\pi$ and $\hat{P}_\pi$ are Markov operators. In particular, they have bounded operator norms $\|P_\pi\|_{\mathrm{op}} \leq 1$ and $\|\hat{P}_\pi\|_{\mathrm{op}} \leq 1$. Let $\beta$ be the discount factor. Finally, let $r_\pi$ be the expected cost function corresponding to policy $\pi$.

Recall from Bertsekas (2011) that the stationary equation for the infinite horizon total value of a policy.

$$V_\pi = r_\pi + \beta P_\pi V_\pi$$

and its plug in counterpart

$$\hat{V}_\pi = r_\pi + \beta \hat{P}_\pi \hat{V}_\pi.$$

Then we have the following proposition

**Proposition 2.** *Let $\|\cdot\|_{L_2(\nu)}$ denote the $L_2$ norm with respect to a generic measure $\nu$, and assume a bounded expected reward function with sup-norm $\|r\|_\infty$. Then, the offline policy evaluation for a given policy $\pi$ satisfies the following error bound*

$$\mathbb{E}\left[\|V_\pi - \hat{V}_\pi\|_{L_2(\lambda_n^\chi)}\right] \leq \frac{2\sqrt{2}\beta}{(1-\beta)^2}\|r_\pi\|_\infty \mathbb{E}[\mathcal{H}^2(\jmath, \hat{\jmath})].$$

## 6.1 Presence of Distribution Shift

In the field of Machine Learning (ML) and data-driven applications, one of the significant challenges is the change in data distribution between the training and deployment stages, commonly known as distribution shift. Most relevant to our setting is *covariate shift* (Tamang et al., 2025) where the features of the underlying model shift between the training and deployment. In this section, we show how our results can be extended to the presence of distribution shifts.

We will start with making appropriate assumptions. Assume that the true data (given by $\{(X_i, a_i, G_i)\}_{i=1}^n$) generating distribution is $\jmath_0$ while for the test dataset (given by $\{(X_i^{(\star)}, a_i^{(\star)}, G_i^{(\star)})\}_{i=1}^m$), has the data generating distribution $\jmath_\star$. That is,

$$\mathbb{P}\left(X_{i+1} \in dy \mid X_i = x, G_i = g, a_i = l\right) = \jmath_0(x, l, y)\mu_\chi(dx) \qquad \text{and}$$
$$\mathbb{P}\left(X_{i+1}^{(\star)} \in dy \mid X_i^{(\star)} = x, G_i^{(\star)} = g, a_i^{(\star)} = l\right) = \jmath_\star(x, l, y)\mu_\chi(dx)$$

For the purposes of this section, we will make the simplifying assumption that the controls are Markovian and that the contexts are i.i.d. (as is often the case in practice). Formally, we make the following assumption.

**Assumption 4.** *Let $x, y \in \chi$, $l \in \mathbb{I}$, $g \in \mathcal{G}$. This assumption is stated in parts.*

- *$G_i$ are i.i.d. with distribution $\gamma$ and for $\mathcal{I} \subset \mathbb{I}$, the control distribution satisfies for some policy distribution $\pi$*

$$\mathbb{P}(a_i \in dl \mid X_i = x, G_i = g) = \mathbb{P}(a_i^{(\star)} \in dl \mid X_i^{(\star)} = x, G_i^{(\star)} = g) = \pi(x, l, g)\mu_{\mathbb{I}}(dl)$$

*Observe that the previous assumption implies that $(X_i, a_i, G_i)$ jointly forms a Markov chain with transition density $\jmath\pi\gamma$ and $(X_i^{(\star)}, a_i^{(\star)}, G_i^{(\star)})$ does the same with transition density $\jmath_\star\pi\gamma$. Let $\nu$ and $\nu_\star$ be the corresponding invariant distributions. Our second assumptions will be on the gap of the distribution shift.*

- *We assume that the distance of the distribution shift is non-negative in the Hellinger metric. Formally,*

$$\hbar(\jmath_\star) := \int \left[\left(\sqrt{\jmath_0} - \sqrt{\jmath_\star}\right)^2 \nu_\star\right] \mu_\chi \mu_\chi \mu_{\mathbb{I}} \mu_{\mathcal{G}} > 0$$

*where we have suppressed the arguments of the functions for notational convenience.*

We briefly discuss the previous assumption. The first part of Assumption 4 is to simplify the analysis, whereas the second part assumes a baseline signal in the shift of the distribution. Note that a distribution shift can both be stationary and non-stationary. The distribution shift is said to be stationary if $\nu = \nu_\star$. Since invariant distributions are not unique to transition kernels this can happen even if $\jmath_0 \neq \jmath_\star$. The distribution shift is said to be non-stationary if $\nu \neq \nu_\star$. We derive our key result under non-stationarity and stationary serves as a special case.

**Proposition 3.** *Assume the conditions of Assumptions 1 and 4. Then, one has the selected estimator $\hat{\jmath}$ from eq. (3.3) satisfying*

$$\mathcal{C}\mathbb{E}\left[\mathcal{H}^2(\jmath_\star, \hat{\jmath})\right] \leq \mathbb{E}\left[\inf_{f \in \mathcal{S}}\left\{\mathcal{H}^2(\jmath, f) + L\frac{\Delta_\mathcal{S}(f)}{n}\right\}\right] + \hbar(\jmath_\star) + \|\nu - \nu_\star\|_{TV}.$$

**Corollary 3.** *Assume the conditions of Assumptions 1 and 4. Further assume $\nu = \nu_*$ Then, one has the selected estimator $\hat{\jmath}$ from eq. (3.3) satisfying*

$$\mathcal{C}\mathbb{E}\left[\mathcal{H}^2(\jmath_\star, \hat{\jmath})\right] \leq \mathbb{E}\left[\inf_{f \in \mathcal{S}}\left\{\mathcal{H}^2(\jmath, f) + L\frac{\Delta_\mathcal{S}(f)}{n}\right\}\right] + \hbar(\jmath_\star).$$

Corollary 3 can be immediately seen from Proposition 3.

## 6.2 Impact of Distribution Shift on Offline Policy Evaluation

In the case where the data is distributionally shifted policy evaluation is impacted by the shift in essence that obtaining the optimal policy with regards to the value function is dependent on the magnitude of the shift. We present the following result quantifying this rigorously.

**Corollary 4.** *Assume the conditions of Assumptions 1 and 4. Then for the true and estimated value functions $V_\pi$ and $\hat{V}_\pi$, one has*

$$\mathbb{E}\left[\|V_\pi - \hat{V}_\pi\|_{L^2(\lambda_n)}\right] \leq \frac{2\sqrt{2}\beta}{(1-\beta)^2}\|r_\pi\|_\infty \left(\mathbb{E}\left[\inf_{f\in\mathcal{S}}\left\{\mathcal{H}^2(\jmath, f) + L\frac{\Delta_\mathcal{S}(f)}{n}\right\}\right] + \hbar(\jmath_\star) + \|\nu - \nu_\star\|\right).$$

*Proof.* From Proposition 2 we know that

$$\mathbb{E}\left[\|V_\pi - \hat{V}_\pi\|_{L_2(\lambda_n^\chi)}\right] \leq \frac{2\sqrt{2}\beta}{(1-\beta)^2}\|r_\pi\|_\infty \mathbb{E}[\mathcal{H}^2(\jmath, \hat{\jmath})].$$

The proof is then a direct application of Proposition 3. $\qquad\square$

## 7 Numerical Results

In this section, we briefly investigate the empirical performance of the proposed density estimator using three simulation models, which we refer to as the *linear Gaussian-type model*, the *additive Gaussian control model*, and the *multiplicative control model*. In all three settings, $X_k$ denotes the state, $U_k$ denotes the context, and $W_k$ denotes the control. The control is generated according to

$$W_k \mid X_k = x \sim N\big(0, \lambda(x)\big),$$

where $\lambda(x)$ is a sigmoid function of $x$. The three transition models are given by

$$\begin{aligned}
\text{(Model I)} \qquad & X_{k+1} = 0.5X_k + \frac{1+U_k}{4} + W_k, \\
\text{(Model II)} \qquad & X_{k+1} = 0.5(X_k + U_k) + W_k, \\
\text{(Model III)} \qquad & X_{k+1} = \frac{X_k}{50X_k + 1} + X_kU_k + W_k.
\end{aligned}$$

For each model, we simulate trajectories from the corresponding transition mechanism and estimate the associated transition density. The entire procedure is repeated over 50 independent replications for each model with 1000 samples from each. The model class we select is the one for the dyadic histograms (example 1) and the one for the splines (example 2). For dyadic histograms, $\ell$ corresponds to the depth of cuts, whereas for splines, $\ell$ corresponds to the maximum degree of the basis polynomials. The reported results are obtained by averaging the estimation performance $\frac{1}{50}\sum_{replications} \mathcal{H}^2(\jmath, \hat{\jmath})$ as the proxy for $\mathbb{E}[\mathcal{H}^2(\jmath, \hat{\jmath})]$ across 50 runs.

## 8 Conclusions

We propose a general estimator for the transition functions of contextual MDPs on continuous state spaces and exhibit finite sample oracle bounds in the randomised Hellinger and deterministic $L_2$ metrics. In addition, we demonstrate the effectiveness of the estimator by theoretically showing deriving cost optimality for the corresponding empirical cost. This gives a general framework towards deriving an optimal control with minimal assumptions on the data generating process; thereby acquitting contextual MDPs from its usual flaws of non-stationarity and irregularity. This method is robust and is independent of structure of the MDP involved, and should therefore be considered as an extremely effective tool for inferential purposes on MDPs and should be readily useful in optimal policy determination in offline setting—a rudimentary version of which is the optimal control determination as presented in our work.

| $\ell$ | Model I (H) | Model II (H) | Model III (H) | Model I (S) | Model II (S) | Model III (S) |
|---|---|---|---|---|---|---|
| 1 | 0.037 | 0.291 | 0.389 | 0.651 | 0.639 | 0.654 |
| 2 | 0.012 | 0.172 | 0.256 | 0.479 | 0.486 | 0.441 |
| 3 | 0.011 | 0.068 | 0.167 | 0.267 | 0.282 | 0.227 |
| 4 | 0.012 | 0.050 | 0.118 | 0.112 | 0.120 | 0.096 |
| 5 | 0.012 | 0.056 | 0.103 | 0.044 | 0.047 | 0.040 |
| 6 | 0.012 | 0.056 | 0.075 | 0.020 | 0.021 | 0.018 |
| 7 | 0.012 | 0.055 | 0.049 | 0.009 | 0.010 | 0.008 |
| 8 | 0.012 | 0.055 | 0.044 | 0.005 | 0.006 | 0.006 |
| 9 | 0.012 | 0.055 | 0.043 | 0.005 | 0.005 | 0.011 |
| 10 | 0.012 | 0.055 | 0.044 | 0.006 | 0.005 | 0.025 |

Table 1: Estimation error results for Model I, II, and III across different values of $\ell$. H stands for histogram, S stands for splines.

We also introduce the problem of optimal cost determination along with the optimal control which has applications in multiple fields like healthcare, economics, etc. Again future directions in these areas lie in the inferential aspects of cost functions and optimal policy determination.

**Limitations and future outlooks** The main bottleneck of $T$-estimators developed in the paper is computation. In fact, it is known for the i.i.d. case, the objective function in eq. (3.2) can be computed in $O(ne^d)$ time for dimension $d$ (see Section 3.2.4 Baraud & Birgé (2009) for more details). However, to the best of our knowledge, $T$-estimators are the only estimators capable of accommodating both non-stationarity and irregularity, and are therefore useful despite its shortcomings.

For future work, given $\jmath$ and $\hat{\jmath}$, we can do policy optimisation using approximate Bellman operators. This has been previously used in the offline RL setting to great effect (Li et al., 2022a;c; Banerjee et al., 2025a), but the contextual question remains open. Furthermore, the question of additional structures on the cost function, like those determined by transport maps between policies (as described in Section 1) remain unanswered, and we plan to explore this question in a future work.

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

## A   Proofs

### A.1   Proof of Theorem 2

The proof follows by first observing that

$$
\begin{aligned}
&- C_n(a^*, G) + C_n(\hat{a}^*, G) \\
&= -C_n(a^*, G) + C_n(\hat{a}^*, G) + \hat{C}_n(a^*, G) - \hat{C}_n(a^*, G) + \hat{C}_n(\hat{a}^*, G) - \hat{C}_n(\hat{a}^*, G).
\end{aligned}
$$

Noting that

$$
-\hat{C}_n(a^*.G) + \hat{C}_n(\hat{a}^*, G) \le 0
$$

by definition one has

$$
\begin{aligned}
&- C_n(a^*, G) + C_n(\hat{a}^*, G) \\
&\le -C_n(a^*, G) + C_n(\hat{a}^*, G) + \hat{C}_n(a^*, G) - \hat{C}_n(\hat{a}^*, G).
\end{aligned}
$$

This implies

$$
\begin{aligned}
&\mathbb{E}\left[ \int \left( -C_n(a^*, G) + C_n(\hat{a}^*, G) \right) d\lambda_n^{(2)} \right] \\
&\le \mathbb{E}\left[ \int \left( -C_n(a^*, G) + \hat{C}_n(a^*, G) \right) d\lambda_n^{(2)} \right] + \mathbb{E}\left[ \int \left( -\hat{C}_n(\hat{a}^*, G) + C_n(\hat{a}^*, G) \right) d\lambda_n^{(2)} \right].
\end{aligned}
$$

For the first term note that

$$
\begin{aligned}
&\mathbb{E}\left[ \int \left( -C_n(a^*, G) + \hat{C}_n(a^*, G) \right) d\lambda_n^{(2)} \right] \\
&\le \mathbb{E}\left[ \int \left| -C_n(a^*, G) + \hat{C}_n(a^*, G) \right| d\lambda_n^{(2)} \right] \\
&\le \mathbb{E}\left[ \int \left| \int L(x, y, a^*, G) s(x, y, a^*, G) d\lambda_n^{(1)} - \int L(x, y, a^*, G) \hat{s}(x, y, a^*, G) d\lambda_n^{(1)} \right| d\lambda_n^{(2)} \right].
\end{aligned}
$$

Using Assumption 3, one has

$$
\begin{aligned}
&\mathbb{E}\left[ \int \left( -C_n(a^*, G) + \hat{C}_n(a^*, G) \right) d\lambda_n^{(2)} \right] \\
&\le \mathbb{E}\left[ \int \left| \int L(x, y, a^*, G) s(x, y, a^*, G) d\lambda_n^{(1)} - \int L(x, y, a^*, G) \hat{s}(x, y, a^*, G) d\lambda_n^{(1)} \right| d\lambda_n^{(2)} \right] \\
&\le \|L\|_\infty \mathbb{E}\left[ \int \int |s(x, y, a^*, G) - \hat{s}(x, y, a^*, G)| \, d\lambda_n \right] \\
&\stackrel{(i)}{=} \|L\|_\infty \mathbb{E}\left[ \int \int |s(x, y, a, G) - \hat{s}(x, y, a, G)| \, d\lambda_n \right] \\
&\le \|L\|_\infty \mathbb{E}\left[ \int \int \left| \sqrt{s(x, a, G, y)} + \sqrt{\hat{s}(x, a, G, y)} \right| \left| \sqrt{s(x, a, G, y)} - \sqrt{\hat{s}(x, a, G, y)} \right| d\lambda_n \right] \\
&\le \|L\|_\infty \left( \mathbb{E}\left[ \int \int \left| \sqrt{s(x, a, G, y)} + \sqrt{\hat{s}(x, a, G, y)} \right|^2 d\lambda_n \right] \right)^{1/2} \left( \mathbb{E}\left[ \int \int \left| \sqrt{s(x, a^*, G, y)} - \sqrt{\hat{s}(x, a^*, G, y)} \right|^2 d\lambda_n \right] \right)^{1/2} \\
&\le \|L\|_\infty \sqrt{2} \left( \mathbb{E}\mathcal{H}(s, \hat{s}) \right)^{1/2}.
\end{aligned}
$$

$(i)$ follows from the linearity assumption in Assumption 3. Since $s \in \mathcal{S}$, using Corollary 1, we have

$$
\mathbb{E}\left[ \int \left( -C_n(a^*, G) + \hat{C}_n(a^*, G) \right) d\lambda_n^{(2)} \right] \le \|L\|_\infty \sqrt{2} \frac{(1 + \dim(s))}{\mathcal{C}} \sqrt{\frac{\log n}{n}}.
$$

We can similarly show that

$$
\mathbb{E}\left[ \int \left( -\hat{C}_n(\hat{a}^*, G) + C_n(\hat{a}^*, G) \right) d\lambda_n^{(2)} \right] \le \|L\|_\infty \sqrt{2} \frac{(1 + \dim(s))}{\mathcal{C}} \sqrt{\frac{\log n}{n}}
$$

by an identical argument.

### A.2 Proof of Theorem 1

This section is dedicated towards the proof of Theorem 1. Our first objective will be to prove that with probability at least $1 - e^{-n\xi}$

$$\mathcal{C}\mathcal{H}^2(\jmath, \hat{\jmath}) \leq \mathcal{H}^2(\jmath, f) + L\frac{\Delta_{\mathcal{S}}(f)}{n} + \xi. \tag{A.1}$$

To that end, we consider two cases.

**Case I** $\left(T(f, \hat{\jmath}) + L\frac{\Delta_{\mathcal{S}}(f)}{n} - L\frac{\Delta_{\mathcal{S}}(\hat{\jmath})}{n} \geq 0\right)$: Let $\alpha = \frac{\sqrt{2}-1}{2\sqrt{2}}$ and $\varepsilon = (2 + 3\sqrt{2})/8$. Since $\alpha < (1 - \varepsilon)$, it follows under this case that,

$$\alpha\mathcal{H}^2(\jmath, \hat{\jmath}) \leq (1 - \varepsilon)\mathcal{H}^2(\jmath, \hat{\jmath}) + T(f, \hat{\jmath}) - L\frac{\Delta_{\mathcal{S}}(\hat{\jmath})}{n} + L\frac{\Delta_{\mathcal{S}}(f)}{n}$$

$$\leq (1 + \varepsilon)\mathcal{H}^2(\jmath, f) + 2L\frac{\Delta_{\mathcal{S}}(f)}{n} + 22\xi. \tag{A.2}$$

**Case II** $\left(T(f, \hat{\jmath}) + L\frac{\Delta_{\mathcal{S}}(f)}{n} - L\frac{\Delta_{\mathcal{S}}(\hat{\jmath})}{n} \leq 0\right)$: To analyse this case we require the following Proposition which is (by now) a standard fare in this literature, and is proved for completeness in Section B.1.

**Proposition 4.** *Set* $\varepsilon = (2 + 3\sqrt{2})/8$. *Under assumptions of Theorem 1, there exists a universal constant* $L_0 > 0$ *such that for all* $L \geq L_0$ *and* $\xi > 0$,

$$\forall f, f' \in \mathcal{S}, \quad (1 - \varepsilon)\mathcal{H}^2(\jmath, f') + T(f, f') - L\frac{\Delta_{\mathcal{S}}(f')}{n} \leq (1 + \varepsilon)\mathcal{H}^2(\jmath, f) + L\frac{\Delta_{\mathcal{S}}(f)}{n} + 22\xi$$

*with probability larger than* $1 - e^{-n\xi}$.

By using the above lemma, with probability larger than $1 - e^{-n\xi}$, for all $f \in \mathcal{S}$,

$$\sup_{f' \in \mathcal{S}} \left\{ (1 - \varepsilon)\mathcal{H}^2(\jmath, f') + T(f, f') - L\frac{\Delta_{\mathcal{S}}(f')}{n} \right\} \leq (1 + \varepsilon)\mathcal{H}^2(\jmath, f) + L\frac{\Delta_{\mathcal{S}}(f)}{n} + 22\xi.$$

Therefore,

$$\alpha\mathcal{H}^2(\jmath, \hat{\jmath}) \leq \alpha\mathcal{H}^2(\jmath, \hat{\jmath}) + T(\jmath, \hat{\jmath}) - L\frac{\Delta_{\mathcal{S}}(f)}{n} + L\frac{\Delta_{\mathcal{S}}(\hat{\jmath})}{n}$$

$$\leq \sup_{f' \in \mathcal{S}} \left\{ \alpha\mathcal{H}^2(\hat{\jmath}, f') + T(\hat{\jmath}, f') - L\frac{\Delta_{\mathcal{S}}(f')}{n} \right\} + L\frac{\Delta_{\mathcal{S}}(\hat{\jmath})}{n}$$

$$\overset{(i)}{\leq} \vartheta(\hat{\jmath})$$

$$\overset{(ii)}{\leq} \vartheta(f) + \frac{1}{n}$$

$$\leq \sup_{f' \in \mathcal{S}} \left\{ \alpha\mathcal{H}^2(f, f') + T(f, f') - L\frac{\Delta_{\mathcal{S}}(f')}{n} \right\} + L\frac{\Delta_{\mathcal{S}}(f)}{n} + \frac{1}{n}$$

where $(i)$ follows from eq. (3.2), and $(ii)$ follows from eq. (3.3). Now observe that, with $\nu = (1-\varepsilon)/\alpha - 1 > 0$, for any pair $f, f' \in \mathcal{S}$ with probability at least $1 - e^{-n\xi}$

$$\alpha\mathcal{H}^2(f, f') \leq (1 + \nu^{-1})\mathcal{H}^2(f, \jmath) + \sup_{f' \in \mathcal{S}} \left\{ (1 - \varepsilon)\mathcal{H}^2(\jmath, f') + T(f, f') - L\frac{\Delta_{\mathcal{S}}(f')}{n} \right\} + L\frac{\Delta_{\mathcal{S}}(f)}{n} + \frac{1}{n}$$

$$\overset{(i)}{\leq} (1 + \nu^{-1})\mathcal{H}^2(f, \jmath) + \left[ (1 + \varepsilon)\mathcal{H}^2(\jmath, f) + L\frac{\Delta_{\mathcal{S}}(f)}{n} + 22\xi \right] + L\frac{\Delta_{\mathcal{S}}(f)}{n} + \frac{1}{n}$$

$$\leq (2 + \varepsilon + \nu^{-1})\mathcal{H}^2(f, \jmath) + 2L\frac{\Delta_{\mathcal{S}}(f)}{n} + 22\xi + \frac{1}{n}.$$

where $(i)$ follows from Lemma 4 with probability at least $1 - e^{-n\xi}$.

This leads to

$$\alpha \mathcal{H}^2(s, \hat{s}) \leq 2\alpha \mathcal{H}^2(s, f) + 2\alpha \mathcal{H}^2(f, \hat{s})$$

$$\leq 2(2 + \alpha + \varepsilon + \nu^{-1})\mathcal{H}^2(f, s) + 4L\frac{\Delta_{\mathcal{S}}(f)}{n} + 44\xi + \frac{2}{n}.$$

Recall that $L > 1$ and the penalty satisfies $\Delta_{\mathcal{S}}(f) \geq 1 \ \forall \ f \in \mathcal{S}$. Therefore, with $\alpha/\min\{2(2 + \alpha + \varepsilon + \nu^{-1}), 44\}$ to be $\mathcal{C}$ we have, with probability at least $1 - e^{-n\xi}$, for all $f \in S$,

$$\mathcal{C} \mathcal{H}^2(s, \hat{s}) \leq \mathcal{H}^2(f, s) + L\frac{\Delta_{\mathcal{S}}(f)}{n} + \xi.$$

Integrating both sides by $\xi$ completes the proof of Theorem 1.

### A.3   Proof of Theorem 3

This section is dedicated to the proof of Theorem 3. Our first step will be to use Tsybakov's reduction scheme Tsybakov (2009) on the minimax risk function. Note that for any $\mathcal{M}' \subset \mathcal{M}$,

$$\mathcal{R}_n = \min_{\hat{a}^* \in \mathbb{I}} \max_{\text{mdp}^* \in \mathcal{M}} \mathbb{E}\left[\int C_n(\hat{a}^*, G) - C_n(a^*, G)\lambda_n^{(2)}\right]$$

$$\geq \min_{\hat{a}^* \in \mathbb{I}} \max_{\text{mdp}^* \in \mathcal{M}'} \mathbb{E}\left[\int C_n(\hat{a}^*, G) - C_n(a^*, G)\lambda_n^{(2)}\right].$$

Now, our objective will be to carefully choose a subclass $\mathcal{M}'$. We will choose in order, (i) state and control space, (ii) loss function, (iii) transition probability function.

**State Space and Control Space:** It will be enough to restrict completely to the finite case. Let for $g_0 \in \mathcal{G}$, and $G_i$ be i.i.d. $\delta_{g_0}$. Observe that this implies $\lambda_n^{(2)} = \delta_{g_0}\frac{1}{n}\sum \delta_{l_i}$ and $\lambda_n = \delta_{g_0}\frac{1}{n}\sum_{i=1}^n \delta_{x_i, l_i}\mu_\chi$. We introduce the notation $\mathcal{I} := \{l_1, \ldots, l_n\}$ and assume that $\mathbb{I} = \{1, 2\}$, $\chi = \{1, \ldots, d+1\}$, with $\mu_\chi, \mu_{\mathbb{I}}$ being counting measures.

Consequently,

$$C_n(l, g) = C_n(l) = \frac{1}{n}\sum_{i=0}^{n-1} L(l, g_0, x_i, y_i)s(x_i, l, g_0, y_i)\mathbb{1}[l \in \mathcal{I}]$$

and

$$\hat{C}_n(l, g) = \hat{C}_n(l) = \frac{1}{n}\sum_{i=0}^{n-1} L(l, g_0, x_i, y_i)\hat{s}(x_i, l, g_0, y_i)\mathbb{1}[l \in \mathcal{I}].$$

In what follows, we suppress all dependence on $g_0$ from notations for convenience.

**Loss Function:** We further assume that range of the function $L$ is $\{0, 1\}$. Observe that this does not violate the positivity assumption since one can always make $L$ positive via translation without changing the chosen control. We will further assume that $L(\cdot, x, \cdot) = 0$ is constant for all $x \in \{1, \ldots, d\}$, $L(\cdot, d+1, d+1) = 0$ and 1 elsewhere.

Consequently,

$$C_n(l) = \frac{1}{n}\sum_{i:x_i=d+1, y_i \neq d+1} s(d+1, l, y_i)\mathbb{1}[l \in \mathcal{I}]$$

and

$$\hat{C}_n(l) = \frac{1}{n} \sum_{i:x_i=d+1, y_i \neq d+1} \hat{\jmath}(d+1, l, y_i) \mathbb{1}[l \in \mathcal{I}]$$

**Transition probabilities:** Fix integers $d \geq 2$ and $n \geq 4$. For notational convenience we assume $d$ is even (the odd case follows by a minor modification). Let $p_\star^{(2)} \in (0,1)$ satisfy

$$p_\star^{(2)} < \frac{1}{d+2}.$$

We choose

$$p_\star^{(2)} = p_\star^{(1)} + \frac{15\varepsilon}{d}.$$

For any probability vector

$$\eta^{(l)} = \left(\eta_1^{(l)}, \ldots, \eta_d^{(l)}, p_\star^{(l)}\right) \in \Delta_{d+1}, \qquad \Delta_{d+1} := \left\{x \in \mathbb{R}_+^{d+1} : \sum_{k=1}^{d+1} x_k = 1\right\},$$

consider the Markov chain on the state space $\{1, 2, \ldots, d, d+1\}$ whose transition matrix is

$$\jmath_{\eta^{(l)}}^{(l)} = \begin{pmatrix} p_1^{(l)} & \cdots & p_d^{(l)} & p_\star^{(l)} \\ \vdots & & \vdots & \vdots \\ p_1^{(l)} & \cdots & p_d^{(l)} & p_\star^{(l)} \\ \eta_1^{(l)} & \cdots & \eta_d^{(l)} & p_\star^{(l)} \end{pmatrix}, \qquad \text{where } p_k^{(l)} = \frac{1 - p_\star^{(l)}}{d} \text{ for } k \in [d]. \tag{A.3}$$

That is, from any state $i \in \{1, \ldots, d\}$ the next state is distributed as $\left(p_1^{(l)}, \ldots, p_d^{(l)}, p_\star^{(l)}\right)$, whereas for state $d+1$ the chain transitions according to $\left(\eta_1^{(l)}, \ldots, \eta_d^{(l)}, p_\star^{(l)}\right)$. For the ease of analysis, we assume that we have access to $M_{\eta^{(2)}}^{(2)}$ with $\eta_k^{(2)} = (1 - p_\star^{(2)})/d$ for all $k \in [d]$.

By checking stationarity conditions, it is straightforward to verify that the stationary distribution of $M_{\eta^{(l)}}^{(l)}$ is $\pi^{(l)} = \pi^{(l)}(\eta^{(l)})$ where

$$\pi_k^{(l)} = \frac{(1 - p_\star^{(l)})^2}{d} + \eta_k^{(l)} p_\star^{(l)}, \qquad k \in [d], \qquad \text{and} \qquad \pi_{d+1}^{(l)} = p_\star^{(l)}. \tag{A.4}$$

Let $(X_1^{(l)}, \ldots, X_m^{(l)}) \sim (M_{\eta^{(l)}}^{(l)}, p^{(l)})$ denote a trajectory of length $m$ started from the initial distribution

$$p^{(l)} = \left(p_1^{(l)}, \ldots, p_d^{(l)}, p_\star^{(l)}\right).$$

Let

$$\sigma = (\sigma_1, \ldots, \sigma_{d/2}) \in \{-1, 1\}^{d/2}$$

and define a parameter vector $\eta^{(1)}(\sigma) \in \Delta_{d+1}$ by

$$\eta^{(1)}(\sigma) = \left(\frac{1 - p_\star^{(1)} + 16\sigma_1\varepsilon}{d}, \frac{1 - p_\star^{(1)} - 16\sigma_1\varepsilon}{d}, \ldots, \frac{1 - p_\star^{(1)} + 16\sigma_{d/2}\varepsilon}{d}, \frac{1 - p_\star^{(1)} - 16\sigma_{d/2}\varepsilon}{d}, p_\star^{(1)}\right),$$

where $\varepsilon$ is small enough so that $\eta^{(l)}(\sigma)$ is a valid probability vector. The truth will be given by

$$\eta_0^{(l)} = \left(\frac{1 - p_\star^{(l)}}{d}, \ldots, \frac{1 - p_\star^{(l)}}{d}, p_\star^{(l)}\right), \qquad M_0^{(l)} := M_{\eta_0^{(l)}}^{(l)}$$

for $l = 1, 2$. Note that this choice is consistent with the stationary distribution as defined in (A.4). We require the following general result on risk lower bounds for estimating Markovian transition matrices which is derived from Wolfer et al. (2021) by observing (see the last line in page 544) that upon $\Sigma$, the minimax estimation problem satisfies

$$\min_{\hat{\jmath}^{(l)}} \max_{\sigma \in \Sigma} \mathbb{P}\left(\|\hat{\jmath}^{(l)} - \jmath^{(l)}\|_1 > 16\varepsilon\right) \geq \Omega\left(1 - n\varepsilon^2\right). \tag{A.5}$$

We are interested in showing only the rate optimality and not those of the associated constants; and have thus suppressed all parameters from the notation in eq. (A.5). Observe that, any estimator we consider will necessarily be chosen uniformly from the class of transition matrices $\jmath^{(l)}_{\eta^{(l)}(\sigma)}$.

**Control Selection:** We can finally write down the costs. Observe that under the current setting

$$\hat{C}_n(2) = C_n(2) = \frac{1}{n}|\{i : x_i = d+1, y_i \neq d+1\}|\frac{1 - p_\star^{(2)}}{d}\mathbb{1}[2 \in \mathcal{I}]$$

since $\hat{\jmath} = \jmath$ in the case $l = 2$ and

$$\hat{C}_n(1) = \frac{1}{n}\left(|\{^{i:x_i=d+1,}_{y_i \neq d+1}\}|\frac{1 - p_\star^{(1)}}{d} + \sum_{\{^{i:x_i=d+1,}_{y_i \neq d+1}\}} \frac{(-1)^{\mathbb{1}[y_i \text{ is odd}]}16\sigma_{y_i}\varepsilon}{d^2}\right)\mathbb{1}[1 \in \mathcal{I}]$$

$$= \frac{1}{n}\left(|\{^{i:x_i=d+1,}_{y_i \neq d+1}\}|\left(\frac{1 - p_\star^{(2)}}{d} + \frac{15\varepsilon}{d^2}\right) + \sum_{\{^{i:x_i=d+1,}_{y_i \neq d+1}\}} \frac{(-1)^{\mathbb{1}[y_i \text{ is odd}]}16\sigma_{y_i}\varepsilon}{d^2}\right)\mathbb{1}[1 \in \mathcal{I}]$$

where $\sigma_{y_i}$ is the $\sigma$ corresponding to $y_i$. Whereas

$$C_n(2) = \frac{1}{n}|\{i : x_i = d+1, y_i \neq d+1\}|\frac{1 - p_\star^{(2)}}{d}\mathbb{1}[2 \in \mathcal{I}]$$

and

$$C_n(1) = \frac{1}{n}|\{i : x_i = d+1, y_i \neq d+1\}|\frac{1 - p_\star^{(1)}}{d}\mathbb{1}[1 \in \mathcal{I}]$$

with the optimal control being 2, and the cost of making a mistake being $\frac{16\varepsilon}{d^2}$. In other words,

$$\int (C_n(1, G) - C_n(2, G))\lambda_n^{(2)} \geq \frac{16\varepsilon}{d^2}.$$

**Probability of Mistake:** To translate an estimation error in the transition matrix to an error in the choice for the optimal control, one needs to ensure

$$\hat{C}_n(1) < \hat{C}_n(2)$$

which happens if $\sum(-1)^{\mathbb{1}[y_i \text{ is odd}]}\sigma_{y_i} \geq 1$. Since the minimax risk bound ensures that the estimator for $\jmath^{(1)}$ shall be chosen randomly among $\jmath^{(1)}_{\eta^{(1)}}$ as defined in (A.3), this is akin to choosing whether there are more negative $\sigma_{y_i}$'s than positive $\sigma_{y_i}$'s. Else, by the definition of $\jmath^{(1)}_{\eta^{(1)}}$, one shall have $\hat{C}_n(1) > \hat{C}_n(2)$. Also, since $\sigma$'s are given values $\pm 1$ at random ($(-1)^{\mathbb{1}[y_i \text{ is odd}]}$ does not matter here since they deterministically flip the sign). This has a probability

$$\sum_{j=0}^{\{i:x_i=d+1,y_i \neq d+1\}/2-1} \binom{\{i : x_i = d+1, y_i \neq d+1\}}{j}\left(\frac{1}{2}\right)^{\{i:x_i=d+1,y_i \neq d+1\}} \geq \frac{1}{4}$$

for large enough $d$. Note that we have no data then choosing the control reduces to a random coin flip with half probability of selecting the incorrect control; thus lower bounding the probability at $1/2$. This is higher than $1/4$ and hence our lower bound for this event captures the no data setting.

**Final Calculations:** We now have everything to prove the lower bound.

$$
\mathbb{E}\left[\int C_n(\hat{a}^*, G) - C_n(a^*, G)\lambda_n^{(2)}\right]
$$

$$
\geq \mathbb{E}\left[\mathbb{1}[\{|{\jmath} - \hat{\jmath}|_1 > 16\varepsilon\}\bigcap\{\sum(-1)^{\mathbb{1}[y_i \text{ is odd}]}\sigma_{y_i} \leq -1]\left(\int C_n(\hat{a}^*, G) - C_n(a^*, G)\lambda_n^{(2)}\right)\right]
$$

$$
= \frac{16\varepsilon}{d}\mathbb{P}\left(\{|{\jmath} - \hat{\jmath}|_1 > 16\varepsilon\}\bigcap\{\sum(-1)^{\mathbb{1}[y_i \text{ is odd}]}\sigma_{y_i} \leq -1\right)
$$

Therefore,

$$
\min_{\hat{a}^*\in\mathbb{I}}\max_{\mathrm{mdp}^*\in\mathcal{M}'}\mathbb{E}\left[\int C_n(\hat{a}^*, G) - C_n(a^*, G)\lambda_n^{(2)}\right]
$$

$$
\geq \frac{16\varepsilon}{d}\min_{\hat{a}^*\in\mathbb{I}}\max_{\mathrm{mdp}^*\in\mathcal{M}'}\mathbb{P}\left(\{|{\jmath} - \hat{\jmath}|_1 > 16\varepsilon\}\bigcap\{\sum(-1)^{\mathbb{1}[y_i \text{ is odd}]}\sigma_{y_i} \leq -1\right)
$$

$$
= \frac{16\varepsilon}{d^2}\min_{\hat{a}^*\in\mathbb{I}}\max_{\mathrm{mdp}^*\in\mathcal{M}'}\mathbb{P}\left(|{\jmath} - \hat{\jmath}|_1 > 16\varepsilon\right)\mathbb{P}\left(\sum(-1)^{\mathbb{1}[y_i \text{ is odd}]}\sigma_{y_i} \leq -1 \mid |{\jmath} - \hat{\jmath}|_1 > 16\varepsilon\right)
$$

$$
= \frac{16\varepsilon}{d^2} \times \frac{1}{4} \times \Omega(1 - n\varepsilon^2).
$$

Setting $\varepsilon = 1/\sqrt{n}$, we have

$$
\min_{\hat{a}^*\in\mathbb{I}}\max_{\mathrm{mdp}^*\in\mathcal{M}'}\mathbb{E}\left[\int C_n(\hat{a}^*, G) - C_n(a^*, G)\lambda_n^{(2)}\right] \geq \Omega\left(\frac{1}{\sqrt{n}}\right).
$$

We then have the minimax risk

$$
\mathcal{R}_n \geq \Omega\left(\sqrt{\frac{1}{n}}\right).
$$

### A.4  Proofs for Offline Policy Evaluation Results

*Proof of Proposition 2.* Since $V_\pi$ and $\hat{V}_\pi$ satisfy the Bellman evaluation equations

$$
V_\pi = r_\pi + \beta P_\pi V_\pi, \qquad \hat{V}_\pi = r_\pi + \beta\hat{P}_\pi\hat{V}_\pi,
$$

subtracting yields

$$
V_\pi - \hat{V}_\pi = \beta P_\pi(V_\pi - \hat{V}_\pi) + \beta(P_\pi - \hat{P}_\pi)\hat{V}_\pi.
$$

Equivalently,

$$
(I - \beta P_\pi)(V_\pi - \hat{V}_\pi) = \beta(P_\pi - \hat{P}_\pi)\hat{V}_\pi,
$$

and since $\|I - \beta P_\pi\|_{\mathrm{op}} > 0$, we can take the operator inverse as,

$$
V_\pi - \hat{V}_\pi = \beta(I - \beta P_\pi)^{-1}(P_\pi - \hat{P}_\pi)\hat{V}_\pi.
$$

Taking $L_2(\lambda_n^\chi)$-norms gives

$$
\|V_\pi - \hat{V}_\pi\|_{L^2(\lambda_n^\chi)} \leq \beta\,\|(I - \beta P_\pi)^{-1}\|_{\mathrm{op}}\,\|(P_\pi - \hat{P}_\pi)\hat{V}_\pi\|_{L^2(\lambda_n^\chi)}.
$$

By the Neumann series bound,

$$
\|(I - \beta P_\pi)^{-1}\|_{\mathrm{op}} \leq \sum_{t=0}^{\infty}\beta^t\|P_\pi\|_{\mathrm{op}}^t = \frac{1}{1 - \beta\|P_\pi\|_{\mathrm{op}}},
$$

since $\beta\|P_\pi\|_{\mathrm{op}} < 1$. It remains to bound $\|(P_\pi - \hat{P}_\pi)\hat{V}_\pi\|_{L_2(\lambda_n^\chi)}$. Note that

$$\|fg\|_{L_2(\lambda_n^\chi)} = \sum f(X_i)g(X_i) \le \sqrt{\left(\sum f^2(X_i)\right)\left(\sum g^2(X_i)\right)} = \sqrt{\|f^2\|_{L_2(\lambda_n^\chi)}\|g^2\|_{L_2(\lambda_n^\chi)}}.$$

Therefore,

$$\begin{aligned}
\|(P_\pi - \hat{P}_\pi)\hat{V}_\pi\|^2_{L_2(\lambda_n^\chi)} &= \|((P_\pi)^{1/2} - (\hat{P}_\pi)^{1/2})((P_\pi)^{1/2} + (\hat{P}_\pi)^{1/2})\hat{V}_\pi\|^2_{L_2(\lambda_n^\chi)} \\
&\le \|((P_\pi)^{1/2} + (\hat{P}_\pi)^{1/2})\hat{V}_\pi\|_{L_2(\lambda_n^\chi)}\|((P_\pi)^{1/2} - (\hat{P}_\pi)^{1/2})\|_{L_2(\lambda_n^\chi)}.
\end{aligned}$$

Since $(\sqrt{a} + \sqrt{b})^2 \le 2(a+b)$, we have

$$\begin{aligned}
\|((P_\pi)^{1/2} + (\hat{P}_\pi)^{1/2})\hat{V}_\pi\|_{L_2(\lambda_n^\chi)} &\le \|\hat{V}\|_\infty \sqrt{2}\|P_\pi + \hat{P}_\pi\|_{L_2(\lambda_n^\chi)} \\
&= 2\sqrt{2}\|\hat{V}\|_\infty.
\end{aligned}$$

Next, since $\hat{V}_\pi = r_\pi + \beta\hat{P}_\pi\hat{V}_\pi$ and $\hat{P}_\pi$ is a Markov operator,

$$\|\hat{V}_\pi\|_\infty \le \|r_\pi\|_\infty + \beta\|\hat{P}_\pi\hat{V}_\pi\|_\infty \le \|r_\pi\|_\infty + \beta\|\hat{V}_\pi\|_\infty.$$

Hence

$$\|\hat{V}_\pi\|_\infty \le \frac{\|r_\pi\|_\infty}{1 - \beta}.$$

The proof is now complete by observing that $\|((P_\pi)^{1/2} - (\hat{P}_\pi)^{1/2})\|_{L_2(\lambda_n^\chi)} = \mathcal{H}^2(s, \hat{s})$. $\qquad\square$

*Proof of Proposition 3.* We first establish that if $\nu^* = \nu$, that is stationarity is satisfied, then the proof is a straightforward application of triangle inequality. Observe that $\mathcal{H}^2(s_\star, \hat{s}) \le 2\mathcal{H}^2(s, \hat{s}) + 2\mathcal{H}^2(s, s_\star)$. It follows from an application of Theorem 1 that

$$C\mathbb{E}\big[\mathcal{H}^2(s_\star, \hat{s})\big] \le \mathbb{E}\left[\inf_{f \in \mathcal{S}}\left\{\mathcal{H}^2(s, f) + L\frac{\Delta_{\mathcal{S}}(f)}{n}\right\}\right].$$

Since $\nu = \nu_\star$, we observe that $\mathbb{E}[\mathcal{H}^2(s, s_\star)] = \int\left[\left(\sqrt{s_0} - \sqrt{s_\star}\right)^2\nu\right]\mu_\chi\mu_\chi\mu_\mathbb{I}\mu_\mathcal{G} = \hbar(s_\star)$. Taking expectation on $\mathcal{H}^2(s_\star, \hat{s})$, and using the previous bounds, we now have the result. Thus, to establish the result, we only need to show that

$$\int\left[\left(\sqrt{s_0} - \sqrt{s_\star}\right)^2\nu\right]\mu_\chi\mu_\chi\mu_\mathbb{I}\mu_\mathcal{G} \le \int\left[\left(\sqrt{s_0} - \sqrt{s_\star}\right)^2\nu_\star\right]\mu_\chi\mu_\chi\mu_\mathbb{I}\mu_\mathcal{G} + \|\nu - \nu_\star\|_{TV}.$$

We observe the following facts

$$\int\left[\left(\sqrt{s_0} - \sqrt{s_\star}\right)^2\right]\mu_\chi \le 2 \text{ and } \nu = \nu_\star + \nu - \nu_\star.$$

Therefore,

$$\int\left[\left(\sqrt{s_0} - \sqrt{s_\star}\right)^2\nu\right]\mu_\chi\mu_\chi\mu_\mathbb{I}\mu_\mathcal{G} = \int\left[\left(\sqrt{s_0} - \sqrt{s_\star}\right)^2\nu_\star\right]\mu_\chi\mu_\chi\mu_\mathbb{I}\mu_\mathcal{G} + \int\left[\left(\sqrt{s_0} - \sqrt{s_\star}\right)^2(\nu - \nu_\star)\right]\mu_\chi\mu_\chi\mu_\mathbb{I}\mu_\mathcal{G}$$

Observe that

$$\begin{aligned}
\int\left[\left(\sqrt{s_0} - \sqrt{s_\star}\right)^2(\nu - \nu_\star)\right]\mu_\chi\mu_\chi\mu_\mathbb{I}\mu_\mathcal{G} &\le \int_{\nu - \nu_\star > 0}\left[\left(\sqrt{s_0} - \sqrt{s_\star}\right)^2(\nu - \nu_\star)\right]\mu_\chi\mu_\chi\mu_\mathbb{I}\mu_\mathcal{G} \\
&\le 2\int_{\nu - \nu_\star > 0}(\nu - \nu_\star) \\
&\le 2\|\nu - \nu_\star\|_{TV}
\end{aligned}$$

$\qquad\square$

## A.5 Proof of Corollary 1

*Proof.* Recall from the introduction that $\mathbb{X} = \chi \times \mathbb{I} \times \mathcal{G} \times \chi$ and let $\mu_{\mathbb{X}}$ be a measure on $\mathbb{X}$ formed by taking the canonical products of $\mu_{\chi}, \mu_{\mathbb{I}}, \mu_{\mathcal{G}}$ etc. Let $L^2(\mathbb{X}, \mu_{\mathbb{X}})$, be the space of all square–integrable functions on $\mathbb{X}$ with respect to the product measure $\mu_{\mathbb{X}}$ according to the natural $L_2$ metric

$$d^2(f, f') = \int_A \left(f - f'\right)^2 d\mu_{\mathbb{X}}.$$

Observe that the space of all models $\mathcal{S} \subset L^2(\mathbb{X}, \mu_{\mathbb{X}})$. We have assumed that $\mathcal{S}$ has a finite dimension, and thus admits a canonical basis expansion. Let that orthonormal basis be given by $f_1^{(b)}, f_2^{(b)}, \ldots$. Observe that, there is no restriction on $L^2(\mathbb{X}, \mu_{\mathbb{X}})$ to be finite dimensional.

We require the following approximation Lemma. A version of this Lemma appears in Sart (2014), who refer to Lemma 5 in Birgé (2006) for its proof, whose proof, in turn, points to Lemma 2 in Birgé & Massart (1998), which is where we also refer the reader for proof.

**Lemma 5.** *For any $f \in \mathcal{S}$, let the basis expansion of $f$ be given by $\{f_1^{(b)}, f_2^{(b)}, \ldots, f_{\dim(f)}^{(b)}\}$ and let*

$$T_f = \left\{ \sum_{i=1}^{\dim f} \alpha_i f_i : \alpha_i \in \frac{2j}{\sqrt{n \dim f}}, j \in \mathbb{Z} \right\}, \quad \mathcal{M}_f = \left\{ g^2 : g \in T_f, d(g, 0) \leq 2 \right\}.$$

*Then, the cardinality of $\mathcal{M}_f$, can be bounded as $|\mathcal{M}_f| \leq (30n)^{\dim(f)/2}$.*

Now let, $\mathcal{M} = \bigcup_g \mathcal{M}_g$, and let $\Delta_{\mathcal{M}}(f) = \inf_{g \in \mathcal{M}_f} \left(\Delta_{\mathcal{S}}(g) + 4 \dim g \log(n)\right)$. Observe that, by an application of Theorem 1 using the penalty $\Delta_{\mathcal{M}}$ and the class of functions to be $\mathcal{M}$, we have :

$$C\mathbb{E}\left[\mathcal{H}^2(\jmath, \hat{\jmath})\right] \leq \mathbb{E}\left[\inf_{f \in \mathcal{M}} \left\{\mathcal{H}^2(\jmath, f) + L\frac{\Delta_{\mathcal{M}}(f)}{n}\right\}\right].$$

Now, an application of Fatou's lemma for exchanging the infimum and the expectation, along with Assumption 2, we get that

$$\mathbb{E}\left[\inf_{f \in \mathcal{M}} \left\{\mathcal{H}^2(\jmath, f) + L\frac{\Delta_{\mathcal{M}}(f)}{n}\right\}\right] \leq \kappa \inf_{f \in \mathcal{M}} \left[d^2(\sqrt{\jmath}, \sqrt{f}) + L\frac{\Delta_{\mathcal{M}}(f)}{n}\right] \tag{A.6}$$

At this point we observe that $d^2(\sqrt{\jmath}, 0) = 1$ (by the virtue of $\jmath$ being a density). Using the triangle inequality $2d^2(\sqrt{\jmath}, 0) + 2d^2(\sqrt{f}, 0) \geq d^2(\sqrt{\jmath}, \sqrt{f})$, it is therefore sufficient to consider $f$ such that $d^2(\sqrt{f}, 0) \leq 2$, since otherwise one can always find a different $f$ such that $d^2(f, \sqrt{\jmath})$ is less than 4. Therefore, one can rewrite the right hand side of the equation in eq. (A.6) as

$$\kappa \inf_{f \in \mathcal{M}} \left[d^2(\sqrt{\jmath}, \sqrt{f}) + L\frac{\Delta_{\mathcal{M}}(f)}{n}\right] = \kappa \inf_{\substack{f \in \mathcal{S} \\ g \in T_f}} \left[d^2(\sqrt{\jmath}, g) + L\frac{\Delta_{\mathcal{M}}(f)}{n}\right]$$

$$\leq \kappa \inf_{f \in \mathcal{S}} \left[\inf_{g \in T_f} d^2(\sqrt{\jmath}, g) + L\frac{\Delta_{\mathcal{S}}(g) + 4\dim g \log(n)}{n}\right]$$

Now, by construction, for all $g \in \mathcal{S}$ one can find $f \in T_f$ such that $d^2(g, f) \leq 1/n$. Therefore,

$$\inf_{g \in T_f} d^2(\sqrt{\jmath}, g) \leq \inf_{g \in \mathcal{S}} d^2(\sqrt{\jmath}, g) + \frac{1}{n}.$$

This completes the proof.

$\square$

## B    Proofs of Auxillary Results

### B.1    Proof of Proposition 4

*Proof.* We need three requisite lemmata. For notational clarity, we introduce two intermediate objects, $\psi(c_1, c_2)$ and $\bar{f}$, defined by

$$\psi(c_1, c_2) := \frac{1}{\sqrt{2}} \frac{\sqrt{c_2} - \sqrt{c_1}}{\sqrt{c_2 + c_1}} \tag{B.1}$$

$$\bar{f}(x, l, y) := \frac{f_1(x, l, y) + f_2(x, l, y)}{2}.$$

**Lemma 6.**

$$\int \psi(f_1, f_2)^2 s \; d\lambda_n \; \leq \; 3\left[ \mathcal{H}^2\left(s, f_2\right) + \mathcal{H}^2\left(s, f_1\right) \right].$$

*Proof.* It is enough to prove

$$\left( \frac{\sqrt{f_2} - \sqrt{f_1}}{\sqrt{\bar{f}}} \right)^2 s \leq 3\left[ \left( \sqrt{s} - \sqrt{f_2} \right)^2 + \left( \sqrt{s} - \sqrt{f_1} \right)^2 \right]$$

after which the proof follows by integrating both sides with respect to $\lambda_n$. This is equivalent to proving

$$\left( \sqrt{f_2} - \sqrt{f_1} \right)^2 s \leq 3\bar{f}\left[ \left( \sqrt{s} - \sqrt{f_2} \right)^2 + \left( \sqrt{s} - \sqrt{f_1} \right)^2 \right].$$

It holds by algebra that $s \leq 2\left[ (\sqrt{s} - \sqrt{\bar{f}})^2 + \bar{f} \right]$. The left hand side can now be rewritten as

$$\left( \sqrt{f_2} - \sqrt{f_1} \right)^2 s \leq 2 \left( \sqrt{f_2} - \sqrt{f_1} \right)^2 \left[ (\sqrt{s} - \sqrt{\bar{f}})^2 + \bar{f} \right]$$

$$= 2\bar{f} \left( \sqrt{f_2} - \sqrt{f_1} \right)^2 \left[ \frac{(\sqrt{s} - \sqrt{\bar{f}})^2}{\bar{f}} + 1 \right]$$

$$= 2\bar{f} \left[ \frac{(\sqrt{s} - \sqrt{\bar{f}})^2}{\bar{f}} \left( \sqrt{f_2} - \sqrt{f_1} \right)^2 + \left( \sqrt{f_2} - \sqrt{f_1} \right)^2 \right] \tag{B.2}$$

Observe that $\left( \sqrt{f_2} - \sqrt{f_1} \right)^2 / \bar{f} \leq (\sqrt{\max\{f_1, f_2\}})^2 / \bar{f}$ which in turn can be upper bounded by 2. Thus,

$$\frac{(\sqrt{s} - \sqrt{\bar{f}})^2}{\bar{f}} \left( \sqrt{f_2} - \sqrt{f_1} \right)^2 \leq 2(\sqrt{s} - \sqrt{\bar{f}})^2$$

$$\leq 2\frac{(\sqrt{f_2} - \sqrt{s})^2 + (\sqrt{f_1} - \sqrt{s})^2}{2},$$

where the second inequality follows from the convexity of the function $x \to (\sqrt{x} - \sqrt{s})^2$ and Jensen's inequality. Since the fact $\left( \sqrt{f_2} - \sqrt{f_1} \right)^2 \leq 2\left[ \left( \sqrt{f_2} - \sqrt{s} \right)^2 + \left( \sqrt{f_1} - \sqrt{s} \right)^2 \right]$ holds algebraically, we now have

$$\frac{(\sqrt{s} - \sqrt{\bar{f}})^2}{\bar{f}} \left( \sqrt{f_2} - \sqrt{f_1} \right)^2 + \left( \sqrt{f_2} - \sqrt{f_1} \right)^2 \leq 3\left[ (\sqrt{f_2} - \sqrt{s})^2 + (\sqrt{f_1} - \sqrt{s})^2 \right].$$

This, when combined with eq. (B.2) completes the proof of our lemma. □

Next, for two functions $f_1$ and $f_2$, define $Z_i$ by

$$Z_i(f_1, f_2) := \psi\left( f_1(X_i, a_i, X_{i+1}), f_2(X_i, a_i, X_{i+1}) \right) - \mathbb{E}[\psi\left( f_1(X_i, a_i, X_{i+1}), f_2(X_i, a_i, X_{i+1}) \right) \mid X_i, a_i]. \tag{B.3}$$

We now state our second lemma:

**Lemma 7.** *Recall from eq. (B.1) that $\varphi(c_1, c_2) = (\sqrt{c_2} - \sqrt{c_1})/\sqrt{2(c_1 + c_2)}$. Then*

$$\left(1 - \tfrac{1}{\sqrt{2}}\right)\mathcal{H}^2\left(\jmath, f_2\right) + T\left(f_1, f_2\right) \;\leq\; \left(1 + \tfrac{1}{\sqrt{2}}\right)\mathcal{H}^2\left(\jmath, f_1\right) + \frac{1}{n}\sum_{i=0}^{n-1} Z_i\left(f_1, f_2\right).$$

*Proof.* The proof of this Lemma share similarities with the proofs of Propositions 2 and 3 in Baraud (2011) or that of Claim B3 in Sart (2014). To begin, observe that it is enough to show

$$\mathcal{H}^2(\jmath, f_2) + T(f_1, f_2) - \mathcal{H}^2(\jmath, f_1) \leq \frac{1}{\sqrt{2}}\left(\mathcal{H}^2(\jmath, f_2) + \mathcal{H}^2(\jmath, f_1)\right) + \frac{1}{n}\sum_{i=0}^{n-1} Z_i(f_1, f_2).$$

Starting from the left hand side, we substitute the expression for $T$ from eq. (3.1), expand all squares, and cancel relevant terms. To be precise, we can write,

$$\text{L.H.S} = \int \left(\sqrt{f_2} - \sqrt{\jmath}\right)^2 d\lambda_n - \int \left(\sqrt{f_1} - \sqrt{\jmath}\right)^2 d\lambda_n + \frac{1}{n}\sum_{i=0}^{n-1} \psi\left(f_1(X_i, a_i, X_{i+1}), f_2(X_i, a_i, X_{i+1})\right)$$

$$+ \int \sqrt{\bar{f}}\left(\sqrt{f_2} - \sqrt{f_1}\right) d\lambda_n + \int (f_1 - f_2)\, d\lambda_n.$$

$$= -2\rho(f_2, \jmath) + 2\rho(f_1, \jmath) + \frac{1}{n}\sum_{i=0}^{n-1} \psi\left(f_1(X_i, a_i, X_{i+1}), f_2(X_i, a_i, X_{i+1})\right)$$

$$+ \int \sqrt{\bar{f}}\left(\sqrt{f_2} - \sqrt{f_1}\right) d\lambda_n$$

$$= -2\rho(f_2, \jmath) + 2\rho(f_1, \jmath) + \frac{1}{n}\sum_{i=0}^{n-1} Z_i(f_1, f_2) + \int \psi(f_1, f_2)\, \jmath\, d\lambda_n + \int \sqrt{\bar{f}}\left(\sqrt{f_2} - \sqrt{f_1}\right) d\lambda_n$$

All that is now left to show is

$$-2\rho(f_2, \jmath) + 2\rho(f_1, \jmath) + \int \psi(f_1, f_2) d\lambda_n + \int \sqrt{\bar{f}}\left(\sqrt{f_2} - \sqrt{f_1}\right) d\lambda_n$$

can be bounded above by $0.5^{0.5}\left(\mathcal{H}^2(\jmath, f_2) + \mathcal{H}^2(\jmath, f_1)\right)$. As before, we start with the left hand side and observe that

$$-2\rho(f_2, \jmath) + 2\rho(f_1, \jmath) + \int \psi(f_1, f_2)\, \jmath\, d\lambda_n + \int \sqrt{\bar{f}}\left(\sqrt{f_2} - \sqrt{f_1}\right) d\lambda_n$$

$$= \int \left[-2\sqrt{f_2 \jmath} + 2\sqrt{f_1 \jmath} + \frac{\sqrt{f_2} - \sqrt{f_1}}{\sqrt{\bar{f}}}\jmath + \sqrt{\bar{f}}\left(\sqrt{f_2} - \sqrt{f_1}\right)\right] d\lambda_n$$

$$= \int \left[\sqrt{\frac{f_2}{\bar{f}}}\left(\sqrt{\bar{f}} - \sqrt{\jmath}\right)^2 - \sqrt{\frac{f_1}{\bar{f}}}\left(\sqrt{\bar{f}} - \sqrt{\jmath}\right)^2\right] d\lambda_n$$

$$\leq \int \sqrt{\frac{f_2}{\bar{f}}}\left(\sqrt{\bar{f}} - \sqrt{\jmath}\right)^2 d\lambda_n$$

$$\leq \sqrt{2}\mathcal{H}^2(\bar{f}, \jmath).$$

The first inequality follows trivially. The second inequality follows from the fact that $f_2/\bar{f} \leq 2$. Now, observe that the function $x \to (\sqrt{x} - \sqrt{\jmath})^2$ is convex in $x$ when $x > 0$. Therefore, using Jensen's inequality, we can write $\sqrt{2}\mathcal{H}^2(\bar{f}, \jmath) \leq \left[\mathcal{H}^2(f_1, \jmath) + \mathcal{H}^2(f_2, \jmath)\right]/\sqrt{2}$. This completes the proof. □

Finally, we adopt from Sart (2014) (see also (Massart, 2007, Chapter 2)) the following iteration of Bernstein's inequality. As before, let $\{\mathcal{F}_0^i\}_{i \geq 0}$ be a filtration and $|g_i| \leq b$ be a bounded random variable adapted to it. Then we have the following lemma.

**Lemma 8.** *Define the sum $\jmath_n := \sum_{i=0}^{n} \left( g_i - \mathbb{E}[g_i|\mathcal{F}_0^i] \right)$ and $V_n := \sum_{i=0}^{n} \mathbb{E}[g_i^2|\mathcal{F}_0^i]$. Then*

$$\mathbb{P}\left( \jmath_n \geq \frac{V_n}{2(\kappa - b)} + x\kappa \right) \leq \exp\left(-x\right) \tag{B.4}$$

*for all $\kappa > b$, and $x > 0$.*

**Proof of Proposition 4:** With Lemmas 6, 7, and 8 in hand, we turn to the proof of Proposition 4. Using $Z_i$ as in eq. (B.3), set $\mathcal{Z}_n = \sum_{i=0}^{n-1} Z_i$ and

$$g_i = \psi\left( f_1(X_i, a_i, X_{i+1}), f_2(X_i, a_i, X_{i+1}) \right).$$

Then, Lemma 8 gives us

$$\mathbb{P}\left( \jmath_n \geq \frac{V_n}{2(\kappa - b)} + x\kappa \right) \leq \exp(-x). \tag{B.5}$$

By rearrangement we reduce $V_n$ to $n \int \psi\left(f_1, f_2\right)^2 \jmath \, d\lambda_n$. Lemma 6 then bounds $\int \psi\left(f_1, f_2\right)^2 \jmath \, d\lambda_n$ by

$$\int \psi\left(f_1, f_2\right)^2 \jmath \, d\lambda_n \leq 3 \left[ \mathcal{H}^2\left(\jmath, f_2\right) + \mathcal{H}^2\left(\jmath, f_1\right) \right].$$

Following eq. (B.5), we get

$$\mathbb{P}\left( \mathcal{Z}_n \geq \frac{3n \left[ \mathcal{H}^2(\jmath, f_2) + \mathcal{H}^2(\jmath, f_1) \right]}{2(\kappa - b)} + x\kappa \right) \leq \exp(-x)$$

which is equivalent to

$$\mathbb{P}\left( \frac{\mathcal{Z}_n}{n} \geq \frac{3 \left[ \mathcal{H}^2(\jmath, f_2) + \mathcal{H}^2(\jmath, f_1) \right]}{2(\kappa - b)} + \frac{x\kappa}{n} \right) \leq \exp(-x). \tag{B.6}$$

By Lemma 7,

$$\left(1 - \tfrac{1}{\sqrt{2}}\right) \mathcal{H}^2\left(\jmath, f_2\right) + T\left(f_1, f_2\right) - \left(1 + \tfrac{1}{\sqrt{2}}\right) \mathcal{H}^2\left(\jmath, f_1\right) \leq \frac{\mathcal{Z}_n}{n}.$$

Substituting this into eq. (B.6) yields, with probability at most $e^{-x}$,

$$\left(1 - \tfrac{1}{\sqrt{2}}\right) \mathcal{H}^2\left(\jmath, f_2\right) + T\left(f_1, f_2\right) - \left(1 + \tfrac{1}{\sqrt{2}}\right) \mathcal{H}^2\left(\jmath, f_1\right) \leq \frac{3 \left[ \mathcal{H}^2(\jmath, f_2) + \mathcal{H}^2(\jmath, f_1) \right]}{2(\kappa - b)} + \frac{x\kappa}{n}.$$

Next, observe that $\psi(\cdot, \cdot) \leq 1/\sqrt{2}$. We set

$$b = 1/\sqrt{2}, \quad x = \frac{\Delta_{\mathcal{S}}(f_1) + \Delta_{\mathcal{S}}(f_2)}{\kappa} + n\zeta, \quad \kappa = \frac{2 + 11\sqrt{2}}{2\sqrt{2} - 2},$$

implying $1.5 \times (\kappa - b) = \left(1 - 1/\sqrt{2}\right)/4$. Hence, with probability at most $\exp\left(-\frac{\Delta_{\mathcal{S}}(f_1) + \Delta_{\mathcal{S}}(f_2)}{\kappa} - n\zeta\right)$,

$$\left(1 - \tfrac{1}{\sqrt{2}}\right) \mathcal{H}^2\left(\jmath, f_2\right) + T\left(f_1, f_2\right) - \left(1 + \tfrac{1}{\sqrt{2}}\right) \mathcal{H}^2\left(\jmath, f_1\right) \leq \frac{1}{4}\left(1 - \tfrac{1}{\sqrt{2}}\right) \left[ \mathcal{H}^2\left(\jmath, f_2\right) + \mathcal{H}^2\left(\jmath, f_1\right) \right] + \frac{x\kappa}{n}.$$

By rearranging terms, substituting the value of $x$, and bounding $\left(1 - 0.5^{0.5}\right) \mathcal{H}^2\left(\jmath, f_1\right)$ by $\left(1 + 0.5^{0.5}\right) \mathcal{H}^2\left(\jmath, f_1\right)$, we conclude

$$\frac{3}{4}\left(1 - \tfrac{1}{\sqrt{2}}\right) \mathcal{H}^2\left(\jmath, f_2\right) + T\left(f_1, f_2\right) \leq \frac{5}{4}\left(1 + \tfrac{1}{\sqrt{2}}\right) \mathcal{H}^2\left(\jmath, f_1\right) + \frac{\Delta_{\mathcal{S}}(f_1) + \Delta_{\mathcal{S}}(f_2)}{\kappa n} + \zeta.$$

Equivalently—using some trivial upper bounds such as $\kappa < 22, 1 < L$ etc., and via some rearrangements—we get

$$(1 - \varepsilon)\mathcal{H}^2(\jmath, f') + T(f, f') - L\frac{\Delta_\mathcal{S}(f')}{n} \leq (1 + \varepsilon)\mathcal{H}^2(\jmath, f) + L\frac{\Delta_\mathcal{S}(f)}{n} + 22\zeta.$$

Now, a union bound implies

$$\mathbb{P}\left(\bigcap_{f,f'\in\mathcal{S}}\left\{(1-\varepsilon)\mathcal{H}^2(\jmath,f') + T(f,f') - L\frac{\Delta_\mathcal{S}(f')}{n} \leq (1+\varepsilon)\mathcal{H}^2(\jmath,f) + L\frac{\Delta_\mathcal{S}(f)}{n} + 22\zeta\right\}\right)$$

$$\leq \sum_{f,f'\in\mathcal{S}}\mathbb{P}\left((1-\varepsilon)\mathcal{H}^2(\jmath,f') + T(f,f') - L\frac{\Delta_\mathcal{S}(f')}{n} \leq (1+\varepsilon)\mathcal{H}^2(\jmath,f) + L\frac{\Delta_\mathcal{S}(f)}{n} + 22\zeta\right)$$

$$\leq \sum_{f,f'\in\mathcal{S}} e^{\left(-\frac{\Delta_\mathcal{S}(f)+\Delta_\mathcal{S}(f')}{\kappa} - n\zeta\right)}$$

$$\overset{(i)}{\leq} \mathcal{C}e^{-n\zeta}$$

where $\mathcal{C}$ is an universal constant; and $(i)$ follows from a combination of Fubini-Tonelli theorem to swap the order of summation of a convergent positive series, and the fact in Assumption 1, that $\sum_f e^{-\Delta_\mathcal{S}(f)} \leq 1$ and $\kappa > 0$. $\qquad\square$

### B.2 Proof of Proposition 1

That $\dim f < \infty$ is obvious. We prove the summability. Observe the two facts that $|\mathcal{M}_\ell| = 2^\ell$, and for all $f \in \mathcal{M}_\ell$, $\Delta_\mathcal{S}(f) = e^a e^{-\ell}$. Therefore,

$$\sum_{f\in\mathcal{S}} e^a e^{-\ell} = e^a \sum_{\ell \geq 1}\left(\frac{2}{e}\right)^\ell$$

$$= e^a \frac{2}{e-2}.$$

Setting $a = \log(\frac{e-2}{2})$ now completes the proof.

