# OpenReview forum: "Adaptive Model Selection in Offline Contextual MDP's without Stationarity"
_TMLR — Accepted by TMLR_

### Review · Reviewer_q8qZ · 2026-02-06

**Summary Of Contributions:**

The paper proposes an estimator for the transition density of contextual Markov decision processes and shows that it satisfies oracle risk bounds without ergodicity assumptions. Finite-sample guarantees for the cost function are also derived via a plug-in approach based on the estimated transition density.

Strengths:
- Extends previous work on transition density estimation for Markov chains using T-estimators, providing oracle risk bounds without ergodicity assumptions.
- Provides finite-sample guarantees for the cost function through a plug-in approach.

Weaknesses:
- The paper does not adequately acknowledge closely related prior work [1].
- The proposed estimator appears to be computationally intractable.


[1] Mathieu Sart, "Estimation of the transition density of a Markov chain", Annales de l’IHP Probabilités et statistiques, vol. 50, pp. 1028–1068, 2014

**Additional Comments:**

I am not an expert in this area, and the paper was difficult to follow. The introduction in particular is not very accessible to readers outside the field and would benefit from clearer exposition and more intuition.

Questions:

1. I guess that any estimator for the transition density can be used in the same way as you use $\hat s$ in Section 5. Moreover, if this new estimator satisfies some theoretical guarantees, these guarantees will be translated to the cost control? A discussion on this is necessary if so.
2. Which is the dependence of the bound of Theorem 2 w.r.t. $\epsilon$?
3. Which is the role of $\delta$ in equation 5.2? Why do you not restrict to $\delta = 1/4$? (because of remark 5)


Typos:
1. The equation in the first contribution I think that is missing an expectation symbol $\mathbb{E}$ on the r.h.s
2. In the proof of Corollary 1 there is a paragraph repeated

**Audience:**

Yes

**Audience Explanation:**

The paper addresses transition density estimation for contextual MDPs with theoretical guarantees that do not rely on ergodicity, which is often unrealistic in practice.

**Broader Impact Concerns:**

No concerns.

**Claims And Evidence:**

No

**Claims Explanation:**

While I have not checked the proofs in full detail, they appear technically correct and largely based on the same techniques as in [1]. The proofs follow similar steps with minor modifications.

However, I have concerns regarding the optimality of the risk bounds (section 4.3). The paper only provides upper bounds on the risk as no minimax lower bounds are given. References are needed to justify claims of optimality.

Moreover, I think the claim "as we prove in Theorem 2, one can use any method to find the optimal control and leverage optimality guarantees endowed by our estimator" is not sufficiently supported. By the use of the word leverage it seems that minimax optimality also holds for guarantees for the cost function. The wording suggests minimax optimality for the cost function, but this is neither formally stated nor proved.

**Requested Changes:**

Most sections (except Section 5) closely parallel [1], which studies an almost identical problem for Markov chains.
The estimator, assumptions, and main theoretical results are essentially the same. To justify acceptance, the authors should more transparently acknowledge this prior work. In particular:

- The estimator on the paper is in essence the same as the one defined in Section 4.1. [1]

- Assumption 2 is the same as Assumption 4.1 in [1].

- Theorem 1 in the paper is in essence the same as Proposition 4.1 in [1].

- Corollary 1 in the paper is in essence the same as Theorem 4.1 in [1].

- The proof techniques of Theorem 1 and Corollary 1 are very similar (if not the same). This should be also acknowledged.

- Corollary 2 in the paper is in essence the same as Corollary 4.3 in [1] (yet this is acknowledged in the paper)


Overall, the novelty relative to [1] should be made explicit and clearly articulated.

---

> ### Author Response · Authors · 2026-03-10
> **Response by Authors**
>
> We thank the reviewer for their careful assessment and are pleased that they view the technical contributions significant. We revised the paper extensively based on various suggestions, and addressed all minor comments in the revised manuscript. We provide detailed responses to their major comments below.
>
> **Weaknesses / Concerns**
>
> **On Relation to Prior Work:** Thank you for pointing this out. We have added citations to [1] at all suggested places throughout the manuscript. That said, we gently note that the estimator s is a penalized minimum-contrast estimator in the sense of Massart (2007), constructed by selecting from a model class in the spirit of the penalized risk framework of Barron et al. (1999). Theorem 1 follows a well-established line of results in the T-estimation literature; see, for example, Baraud (2011), Sart (2014), and Baraud et al. (2017). To the best of our knowledge, these are among the few available methods that can simultaneously accommodate non-stationarity and model irregularity. The main point is that the concentration inequality in Lemma 8, versions of which appear in many prior papers, is very general and continues to be useful for our proofs.
>
> **Computational tractability:** We thank the reviewer for raising this important point and respond in two parts.
>
> We agree that the estimator arising from T-estimation can be computationally demanding, and that this limitation should be stated clearly. In the revised manuscript, we now discuss this issue explicitly. The computational complexity depends on both the dimension of the state space and the size of the model class and typically grows exponentially with these quantities. We now highlight this limitation in the discussion of the method and in the limitations section. In particular, prior to Example 2 we now explicitly state:
>
> The computational cost of computing the estimator is $O(e^{\ell d})$; (as can be seen in Proposition A.1 of Sart (2014)).
>
> We further note that, although the main contribution of the paper is theoretical, the framework still provides a concrete implementation pathway. The procedure consists of selecting a structured model class, computing an estimator $\hat s$ within that class, and then plugging $\hat s$ into the downstream cost or value functional. For low-dimensional problems this can be implemented directly, for example through exhaustive or depth-first search over the admissible sieve class. In our own experiments, the computational cost was minimal in the one-dimensional setting (d=1), which allowed us to include the numerical study in the paper. At the same time, we agree that these experiments do not constitute a full scalability analysis, and the revised manuscript now makes this limitation more explicit.
>
> **Questions**
>
> **Generality with respect to the transition density estimator:** You are entirely correct, and we thank the reviewer for pointing this out. Any transition density estimator can in principle be incorporated into the framework used in Section 5. If that estimator satisfies appropriate theoretical guarantees, these can be translated into corresponding guarantees for cost control. We agree that this point deserved to be made explicit. The revised version of Theorem 2 is intended to address exactly this issue.
>
> **Dependence of the bound in Theorem 2 on $\epsilon$:** We have rewritten Theorem 2 completely and substantially strengthened the argument. The new proof is significantly tighter, and Theorem 2 now establishes the following improved excess-cost bound: for any $\hat a \in \hat a^\star(G)$ and $a\in a^\star(G)$.
> $$E\left[\int \bigl(C_n(\hat a_n,G)-C_n(a,G)\bigr)\,d\lambda_n^{(2)}\right] = O\left(\sqrt{\frac{\log n}{n}}\right).$$
>
> **Role of $\delta$:** As in the response to Q2, Theorem 2 has now been completely revised, and the new bound is independent of $\delta$
>
> **Minimax Optimality for the Cost Function:** In the revised manuscript, we complement this upper bound with a new minimax lower bound in Theorem 3. There, we define the minimax excess-cost risk over contextual MDPs and prove a matching lower bound up to logarithmic factors. Thus, the control result is no longer just an upper bound for a particular estimator, but also a rate-optimality statement showing that the recovered rate cannot generally be improved over this model class. We hope that this addresses the reviewer's concern.
>
> ___We hope we have answered all questions, and our answers help the reviewer to reevaluate the merits of the paper. We are glad to provide further clarifications in the discussions period.___

---

### Review · Reviewer_TJty · 2026-02-21

**Summary Of Contributions:**

Contextual MDPs are hard to specialize to offline datasets due to non-stationarity and model irregularity. This paper tackles the issue by using T-estimation, which has found recent success in general density estimation procedures, and Markovian contexts. They provide an algorithm for selecting an estimator given a sample from a contextual MDP and using it to derive oracle risk bounds under specific loss functions, and provide finite sample guarantees for optimal control with density estimation. Specifically, they prove an oracle inequality in empirical Hellinger loss which is bounded by the smallest Hellinger distance with any other function in the hypothesis class, along with an error that decays on the rate of $\frac{\mathrm{dim}(f)\log n}{n}$. Moreover, for standard smooth classes, they attain the optimal non-parametric rate, and establish some non-asymptotic bounds for the plug-in cost gap between the control selected by the UCB-style surrogate and the true optimal controller. The proof contributions are as follows:

A non-parametric framework for offline contextual MDPs that:
a) adaptively estimates transition density without stationarity/ergodicity assumptions
b) delivers oracle risk bounds in an empirical Hellinger metric,
c) transfers these guarantees to cost minimization and optimal control selection,
providing minimax optimal guarantees (upto log factors), while assuming smoothness.

Strengths:
- Emphasizes the use of T-estimation principle, introduced by Baraud and Birgé (2000) to prove robust performance.
- Statistical assumptions on $f$ and $\Delta_{\mathcal{S}}(f)$ are mild and generally weak

Weakness:
- Proof writing in some places is not really understandable, and I really believe that updating the proof to make it more readable (e.g. I do think the general structure of the logic seems reasonable that it should be correct). One such issue, for instance, is that in A.1 (page 13), there is no “=“ between the first line and second line in the proof.
- No mention of where the constant 22 came from
- In the proof of Lemma 6, in the last line, I think you’d want to be a bit more carefully since you’re working with a double summation which you’d be swapping. You would have to write $e^{-n\zeta} \cdot \sum_f \sum_{f’} e^{-\Delta_S(f_1)/\kappa} e^{-\Delta_S(f_2)/\kappa}$ and use finiteness of $\mathcal{S}$ to show that you can separate/swap the summations (using Fubini-Tonnelli’s theorem)?
- Perhaps the biggest weakness of the work is that it is unclear how practical the result is without any numerical simulations/benchmarks. It would be great if the authors could verify an implementation of their algorithm on any of the standard benchmarks, for e.g. on a simple bandit setting (https://github.com/SarahLiaw/ctx-bandits-mcmc-showdown.git)

Questions for the author:
- they assume standard smoothness regimes but criticize strong smoothness assumptions in previous work. It would be great if they could draw a detailed distinction.
- they say that previous work assumes stationarity, but this is not the case (for instance, nonstationary MDPs have been studied, e.g. Predictive Control and Regret Analysis of Non-Stationary MDP with Look-ahead Information (Zhang et al 2024)), but what does it mean for it to be offline nonstationary? Is it just that the input distribution is not fixed?
- What are the regularity assumptions made on $\mathcal{S}$? They do not appear to be given explicitly,
- Definition 1 (why is this the model seelction procedure the right thing to look at)?
- Proof of proposition 7. Isn’t there a nicer way to bound this by swapping the integral and expectations using Fubini-Tonnelli and using that the expectation of the indicators are probability events? Moreover, how do you have the first bound of the expectation of the integral being at least $(r_S/n)^\delta$?

**Audience:**

No

**Audience Explanation:**

See summary of contributions

**Broader Impact Concerns:**

-

**Claims And Evidence:**

No

**Claims Explanation:**

Generally I think the work has promise and the idea should be right, but I am not convinced by the existing proof (for e.g. look at my comments above) structure and detail/rigor. I think the work could benefit from a significant revision of the presentation of the proofs and by stating the contributions in a more precise/measured manner.

**Requested Changes:**

- page 3, 2nd paragraph in section 2 repeats the word “estimators”
- In page 9, some of the equations are typeset a bit strangely. Step 1/3 the equation can be made into a straight line. Step 2, the conditional expectation should be in one line,
- Proof writing in some places is not really understandable, and I really believe that updating the proof to make it more readable (e.g. I do think the general structure of the logic seems reasonable that it should be correct). One such issue, for instance, is that in A.1 (page 13), there is no “=“ between the first line and second line in the proof.

---

> ### Author Response · Authors · 2026-03-10
> **Response by Authors**
>
> We thank the reviewer for their careful review of the body and the proofs, and are pleased to find that they view the work promising. We revised the paper extensively based on various suggestions by sharpening the excess-cost bound, proving a matching minimax lower bound up to logarithmic factors, and expanding the offline RL connection. We have addressed all minor comments in the revised manuscript, and provide detailed responses to their major comments below.
>
> **Weaknesses**
>
> 1. **On Exposition:** We thank the reviewer for the careful reading and helpful suggestion. In response, we thoroughly revised the paper, correcting typographical errors and notational inconsistencies throughout, and substantially rewrote the relevant proofs for clarity. In particular, we reorganized and streamlined the main proof so the argument proceeds more directly and is easier to follow. We hope these revisions address the reviewer’s concerns and improve the manuscript’s readability.
>
> 2. **On 22:** We thank the reviewer for flagging this omission; $22$ is only a technical constant arising from positivity of a denominator term in the Rio-style Bernstein inequality of Lemma~8, and to avoid unnecessary confusion in the main text, we have replaced it with a universal positive constant $\kappa$.
>
> 3. **On Simulations:**  We appreciate this observation and agree that the earlier draft did not provide enough evidence for the broader discussion of practical relevance. Our intent was not to claim that the present work is a fully benchmarked or directly deployable algorithmic solution, but rather that the theoretical framework accommodates features arising in realistic data settings, including non-stationarity, irregular transition structure, and weak assumptions on the data-generating process. We revised the introduction to state this point more precisely and modestly.
>
> To strengthen the empirical side, we added a new Section 8 with simulation studies. There, we examine the proposed estimator on three contextual MDP data-generating mechanisms: a linear Gaussian-type model, an additive Gaussian control model, and a multiplicative control model. In each setting, we simulate trajectories, estimate the transition density using two sieve families---dyadic histograms and splines---and repeat the full procedure over 50 independent Monte Carlo runs with 1000 samples per run. We report the resulting behavior across model classes and sieve complexity levels $\ell$ to illustrate finite-sample performance.
>
> We view these experiments as an initial empirical complement to the theory. They show that the estimator can be implemented in concrete examples and that the framework yields a numerically executable estimation procedure in controlled contextual settings. At the same time, we do not present them as a definitive assessment of large-scale practical performance. Their role is narrower: to provide initial computational validation and show that the theoretical construction can be instantiated and studied in finite samples.
>
> 4. **On Smoothness Regimes:** We thank the reviwer for this question and gently note that our main theorems (Theorem 1, 2, 3) are all proved without the smoothness assumptions. The smoothness assumptions serve as corollaries to show  that the model selection procedure produces minimax risk when such smoothness assumptions on the density are made. We further note that the main issue with non-parametric estimation is not the smoothness assumption, but the requirement to know *apriori the smoothness parameter to set the bin-size*. This requirement is what is alleviated by adaptive estimation (with a computational tradeoff).
>
> 5. **On Stationarity:** If the data-generating process were allowed to be a fully general non-stationary MDP, the logging policy could change over time and depend on the entire trajectory. This would substantially broaden the model class and complicate both analysis and estimation. Our main contribution in this paper is to demonstrate theoretically that under mild conditions non-stationarity does not pose a significant challenge towards estimation of the transition densities, which is an important object in general Stat/ML tasks.

---

> ### Author Response · Authors · 2026-03-10
> **Response by Authors (2)**
>
> 6. **Regularity on S:** We thank the reviewer for this question and gently remind them that we have explicit examples in Section 4.2. Broadly, the model class is a \textbf{choice} made by the practitioner and any performance guarantees are contingent upon the choice of a rich enough model class. We do agree that a heuristic discussion is beneficial and we have added the following discussion after Remark 3.
>
>  >**On the choice of the model class $\mathcal{S}$.** In practice, the performance of the estimator depends on the complexity of the candidate model class $\mathcal{S}_\ell$. The precise choice is somewhat ambiguous but Larger model classes typically yield more precise estimation, with the obvious tradeoff of under-penalizing bad models for small sample sizes.
> > - For smaller sample sizes, smaller classes may be beneficial to avoid excessive bias with the added risk of under-penalizing if the model class is too large.
> >- For moderate sample sizes, setting standard model classes like those in examples below (see Example 1 or 2) work well.
> >- For larger sample sizes, even richer classes may suffice.
>
> 7. **Rationale for Model Selection:** We thank the reviewer for this question and gently remind them that model selection is a powerful method of non-parametric adaptive density estimation with a rich history and robust theoretical guarantees (even encompassing the recent $\rho$-estimators). To further clarify our motivation we have included the following under equation (3.3) of the revised manuscript
>
> >   Observe that $\hat s$ is precisely the minimum-contrast estimator of Massart (2007) and, following the foundational PTRF paper Barron et al. (1999), is estimated via a penalised selection procedure from a model class. Recently, this estimator has been used with great success in model selection for Markov chains (Sart, 2014), and in the following section we investigate its properties in the contextual MDP setting.
>
> 8. **Proof Improvement:** We thank the reviewer for this important observation and respond in two parts.
>
> The earlier proof has been replaced by a sharper and more direct analysis, in a line similar to what the reviewer suggests. Theorem~2 now establishes the following improved excess-cost bound: for any $\hat a_n \in \hat a^\star(G)$ and $a \in a^\star(G)$,
> $$ E\left[\int \bigl(C_n(\hat a_n,G)-C_n(a,G)\bigr)\,d\lambda_n^{(2)}\right] = O\left(\sqrt{\frac{\log n}{n}}\right).$$
>
> The proof is now considerably tighter and more transparent. Instead of relying on a qualitative plug-in argument, it directly compares the true and estimated cost functionals and controls the difference using the Hellinger-risk bound for transition-density estimation, together with standard norm inequalities and the boundedness of the cost function.
>
> Furthermore, this upper bound is now complemented by a new minimax lower bound in Theorem~3. In that result, we formulate the minimax excess-cost risk over the class of contextual MDPs under consideration and prove a corresponding lower bound that matches the upper rate up to logarithmic factors. As a result, the excess-cost theory is no longer limited to an upper bound for one particular estimator; it is now accompanied by a rate-optimality result showing that, in general, this rate cannot be improved over the model class considered.
>
> ___We hope we have answered all questions, and our answers help the reviewer to reevaluate the merits of the paper. We are glad to provide further clarifications in the discussions period.___

---

> > ### Comment · Reviewer_TJty · 2026-04-07
> > **Response to Authors**
> >
> > I thank the authors for their detailed response.
> >
> > I particularly appreciate the authors' comments on the choice of the model class $S$, and for their constructive responses.
> >
> > I maintain my positive rating for this work.
> >
> > Thanks,
> >
> > Reviewer

---

### Review · Reviewer_rmbW · 2026-03-02

**Summary Of Contributions:**

The paper is mainly a nonparametric adaptive transition-density estimation paper. The target is the conditional distribution of the next state given the current state, action, and context. The main statistical tool is model selection by tests (T-estimation), adapted to the sequential setting of contextual Markov process from trajectory data.

The central theoretical result is an oracle inequality in empirical Hellinger loss. This shows that the estimator performs almost as well as the best model in a candidate collection, up to a complexity penalty. The guarantee does not rely on stationarity or ergodicity assumptions, which is one of the technical motivations of the paper. The authors extend classical T-estimator arguments, which are usually developed for i.i.d. data, to controlled dependent data by using martingale concentration tools. This is the main technical work behind the proofs.

From the oracle inequality, the paper derives adaptive bias–complexity bounds. These results show that the estimator automatically adapts to the unknown complexity of the transition density through model selection penalties. Under smoothness assumptions and histogram-type approximation spaces, the paper obtains near-minimax nonparametric rates (up to logarithmic terms). This connects the results to standard adaptive estimation theory.

The paper also introduces a plug-in one-step control rule. The estimated transition density is used to estimate a one-step cost, and actions are selected by minimizing this estimated cost with an additional penalty term.

Strengths:

The paper brings model-selection-by-tests (rarely used in modern ML) into sequential controlled data settings. Some TMLR readers interested in robust estimation or uncertainty-aware modeling may find this useful.

1. Statistical theory for adaptive density estimation
The oracle inequality and adaptive model-selection framework are legitimate contributions. Extending T-estimation to controlled dependent trajectories using martingale tools is technically interesting and clean.

2. Weak assumptions compared to classical Markov analyses
No stationarity or mixing assumptions. That is meaningful from a theory standpoint because many results in nonparametric Markov estimation rely on stronger conditions.

3. Minimax nonparametric estimation
The paper recovers standard adaptive rates under smoothness assumptions.

Weaknesses:

1. Framing as offline RL / optimal control is a bit surprising
What is actually solved is a one-step plug-in decision problem. In this paper, there are no Bellman operator, no long-horizon value, no policy evaluation under distribution shift for example. Calling this “offline contextual MDP control” feels like a rebranding.

2. Control contribution is weak
The excess-cost bound is slow $n^{-1/4}$ and comes from a generic plug-in argument. There is no modern policy-learning perspective (are there any semiparametric efficient perspective here instead?).

3. No computational story
T-estimators are known to be hard to compute. The paper provides no practical algorithm, scalability analysis, or realistic implementation pathway. For TMLR, this hurts impact.

4. No numerical simulations at all.
This is very surprising for a paper that claims to "close a gap between practice and theory"... Hurts impact for TMLR audience.

5. Little connection to modern literature
No comparison to contemporary or recent offline RL / OPE / model-based RL frameworks. The paper reads more like classical nonparametric statistics relabeled as control.

**Audience:**

No

**Audience Explanation:**

While I believe there is a lot of statistical value in this paper, I am not convinced TMLR is the good venue for this work. I will read other reviewers input and see what the authors can do in terms of improvements wrt to the practical aspects of their method.

**Broader Impact Concerns:**

N.A

**Claims And Evidence:**

No

**Claims Explanation:**

Yes and No.

As per the theoretical contributions of the paper, there are supported by accurate and convincing arguments and proof.
As per the overall claim of the paper to "close a gap between practice and theory", I find it very surprising that no algorithm is cleanly discussed in terms of practical implementations, nor complexity, nor analysis, nor any experiments.

**Requested Changes:**

Discuss practical implementations, discuss experiments, discuss more the algorithm in practice.

---

> ### Author Response · Authors · 2026-03-10
> **Response by Authors**
>
> We thank the reviewer for their careful assessment and are pleased that they view the statistical contribution as substantial. We revised the paper extensively in response to their suggestions, which we believe better situates our work in the broader machine learning literature. More broadly, we strengthened the control contribution by sharpening the excess-cost bound and proving a matching minimax lower bound up to logarithmic factors, and expanded the offline RL connection by adding long-horizon Bellman-based OPE results with explicit guarantees under distribution shift. We address all minor comments and respond to the major suggestions below.
>
> **Weaknesses**
>
> 1. **On Framing:**  We thank the reviewer for this important observation. We agree that the original version underdeveloped the connection to offline RL and framed the main technical results primarily through a plug-in control rule based on one-step transition estimation. In response, we substantially revised the manuscript to make the RL content explicit and clarify the scope of our contribution.
>
> Concretely, we added a new Section 7 connecting the statistical estimation results to standard offline RL questions. In particular, we now introduce the discounted infinite-horizon value function under a stationary policy $\pi$, together with the Bellman equation
> $V^\pi = r^\pi + \beta P^\pi V^\pi$ and its plug-in analogue.
>
> We then show in Proposition 2 that our transition-density estimation bounds imply the quantitative offline policy evaluation guarantee:
> $${E}\bigl[\|V^\pi-\hat V^\pi\|\bigr] \leq O\left(\frac{\beta}{(1-\beta)^2}\,\|r^\pi\|_\infty\,{E}\left[H^2(s,\hat s)\right]\right).$$
>
> This gives an explicit long-horizon offline policy evaluation error bound in the discounted contextual MDP setting.
>
> We also added Proposition 3 and Corollary 4 to address distribution shift. Proposition~3 gives a transition-estimation guarantee under shifted data, with additional terms depending on the Hellinger gap $h(s_\star)$ and the discrepancy between invariant measures $\|\nu-\nu^\star\|_{TV}$. Corollary 4 then propagates this shift directly into offline policy evaluation.
>
> Overall, we believe adaptive estimation is a relatively niche topic in machine learning, and that our work is well aligned with TMLR’s objective of encouraging discussion of less widely studied topics in machine learning.
>
> 2. **On Optimality of the Control:** We thank the reviewer for this important point. We agree that, in the earlier version, the control guarantee appeared to follow from a generic plug-in argument, which made the control contribution seem weaker than intended. In the revised manuscript, we have substantially strengthened this part of the paper in two ways.
>
> Theorem 2 now gives a significantly sharper excess-cost guarantee. Specifically, for any $\hat a_n \in \hat a^\star(G)$ and $a \in a^\star(G)$, we show
>
> $$E\left[\int \bigl(C_n(\hat a_n,G)-C_n(a,G)\bigr)\,d\lambda_n^{(2)}\right]=O\left(\sqrt{\frac {\log n}n}\right),$$
>
> an excess-cost rate of order $O(\sqrt{\log n / n})$. The proof has also been tightened substantially: instead of a loose qualitative plug-in argument, it now directly compares the true and estimated cost functionals and controls the resulting terms using the Hellinger-risk bound for transition-density estimation, standard norm inequalities, and boundedness of the cost function.
>
> We also complement this upper bound with a new minimax lower bound in Theorem~3. There, we define the minimax excess-cost risk over contextual MDPs and prove a matching lower bound up to logarithmic factors. Thus, the control result is no longer just an upper bound for a particular estimator, but also a rate-optimality statement showing that the recovered rate cannot generally be improved over this model class.
>
> Regarding the reviewer’s observation that the method remains plug-in in nature, we agree and emphasize that this is by design. Plug-in procedures have a long history of strong optimality properties in statistical decision problems, and revised Theorems~2 and 3 place the present method in that tradition. To make this clear to readers, we added the following remark in the revised manuscript.
>
> >  We also remark that the minimax risk is achieved by a plug-in estimator which was previously known in finite state-control spaces (Agarwal et al., 2020; Li et al., 2022a). Our results therefore, extend this important body of literature by extending it to compact (but possibly infinite) state-control spaces.
>
> We also agree that a semiparametric-efficiency perspective is interesting, but it is distinct from our nonparametric, minimax goal of adaptively estimating the transition mechanism and resulting optimal-control decision rather than a finite-dimensional smooth functional; we do not know whether our techniques extend to the semiparametric setting, but view this as an important direction for future work.

---

> ### Author Response · Authors · 2026-03-10
> **Response by Authors (2)**
>
> 3. **On Computation:** We thank the reviewer for flagging this important point and respond in two parts.
>
> First, we agree that computation is a genuine limitation of T-estimation, and that this should be stated more clearly for a TMLR audience. In the revised manuscript, we now address this issue explicitly. The computational complexity depends on both the state dimension and the size of the model class and is typically exponential in these quantities. We now state this clearly in the limitations section, noting that even in the i.i.d. setting the objective underlying T-estimation can be costly to evaluate. In addition, before Example 2 we explicitly state that the computational cost of computing the estimator is $O(e^{\ell d})$, and refer the reader to Proposition A.1 of Sart (2014).
>
> Second, although the paper is primarily theoretical, it does provide a concrete implementation pathway: choose a structured model class, compute $\hat s$ over that class, and then plug $\hat s$ into the downstream cost or value functional. For low-dimensional problems, the estimator can be computed naively by exhaustive or depth-search style traversal over the admissible sieve class. In our own experiments, the runtime was modest in dimension $d=1$, which is why we felt comfortable including a numerical study. At the same time, we agree that these experiments do not amount to a scalability analysis, and the revised manuscript now makes this limitation more explicit.
>
> 4. **On Simulations:** We also agree that the original submission did not sufficiently support the claim about bridging ``practice and theory.'' Our intended meaning was not that the paper closes the gap between theoretical analysis and deployed large-scale practice in an algorithmic sense, but rather that it narrows the gap between realistic data features---such as non-stationarity, model irregularity, and weak structural assumptions on logged data---and the assumptions typically imposed in existing theory. We now state this more carefully in the introduction.
>
> That being said, we agree with the importance of simulations to ground the theory empirically, we have added a new Section~8 containing numerical simulations. There, we study the estimator on three simulated contextual MDP models: a linear Gaussian-type model, an additive Gaussian control model, and a multiplicative control model. For each model, we generate trajectories, estimate the transition density over two sieve classes---dyadic histograms and splines---and repeat the procedure over 50 Monte Carlo replications with 1000 samples per run. These experiments show that the estimator can be instantiated on concrete model classes and provide an initial finite-sample validation of the theoretical framework, without claiming a definitive assessment of large-scale practical performance.
>
> 5. **Connections to Contemporary RL:** Finally, we have added stronger connections to contemporary model-based offline RL in the revised manuscript, including Remark 5 and new results such as Propositions 2--3 and Corollary 4. More broadly, the manuscript aims to bring adaptive nonparametric estimation, oracle inequalities, minimax optimality, and robustness to irregular logged data into the theory of offline contextual MDPs, while making clearer contact with modern OPE and model-based RL perspectives.

---

> > ### Author Response · Authors · 2026-03-10
> > **Response by Authors (3)**
> >
> > **Other Questions**
> >
> > 1. **On Empirical Evaluations:** To strengthen the paper on the empirical side, we have added a new Section 8 with simulation studies. We examine the proposed estimator on three contextual MDP data-generating mechanisms: a linear Gaussian-type model, an additive Gaussian control model, and a multiplicative control model. In each setting, we simulate trajectories, estimate the transition density using two sieve families—dyadic histograms and splines—and repeat the full procedure over 50 independent Monte Carlo runs with 1000 samples per run. We report performance across model classes and sieve complexity levels $\ell$ to illustrate finite-sample behavior.
> >
> > We view these experiments as an initial empirical complement to the theory. They show that the estimator can be implemented in concrete examples and that the framework yields a numerically executable procedure in controlled contextual settings. We do not present them as a definitive study of large-scale practical performance; rather, they provide an initial computational validation and complement the expanded discussion of implementation and complexity.
> >
> > 2. **On Suitability:** We thank the reviewer for this assessment and for recognizing the statistical contributions of the work. Our goal is to develop a rigorous statistical framework for learning transition structures and control policies in contextual MDPs under weak structural assumptions. We believe this is closely connected to topics of interest to the TMLR community, and the revised version of the manuscript should be of interest to the broader community working in a variety of machine learning topics like offline reinforcement learning, distribution shift, and policy evaluation under model uncertainty.
> >
> > To address the reviewer’s concern about practical aspects, we have added a new section with simulation studies illustrating implementation in concrete contextual MDP settings, and we have expanded the discussion of implementation, computational considerations, and limitations. Our intention is not to present a fully optimized large-scale algorithm, but rather a principled statistical approach whose guarantees may inform future algorithmic developments.
> >
> > 3. **Practical Changes:** We gently note that the manuscript already contains explicit examples in Section 4.2. At the same time, we agree that additional heuristic discussion is useful for practitioners, and we have therefore added the following discussion after Remark 3.
> >
> >  >**On the choice of the model class $\mathcal{S}$.** In practice, the performance of the estimator depends on the complexity of the candidate model class $\mathcal{S}_\ell$. The precise choice is somewhat ambiguous but Larger model classes typically yield more precise estimation, with the obvious tradeoff of under-penalizing bad models for small sample sizes.
> > > - For smaller sample sizes, smaller classes may be beneficial to avoid excessive bias with the added risk of under-penalizing if the model class is too large.
> > >- For moderate sample sizes, setting standard model classes like those in examples below (see Example 1 or 2) work well.
> > >- For larger sample sizes, even richer classes may suffice.
> >
> >
> > ___We hope we have answered all questions, and our answers help the reviewer to reevaluate the merits of the paper. We are glad to provide further clarifications in the discussions period.___

---

### Author Response · Authors · 2026-03-10
**Overview of the revision**

We thank all reviewers for their detailed feedback and their insightful comments. *We have addressed all reviewer comments in the revised version of the manuscript*. In this overview, we highlight 2 major additions which are in response to feedback by multiple reviewers. **All changes in the revised manuscript are marked by color in blue.**

**On Optimality Of Cost Function:**  In the earlier version, the control guarantee was understated due to the use of loose analytical arguments. In the revised manuscript, we have strengthened this component substantially in two ways.

First, Theorem 2 now gives a significantly sharper excess-cost guarantee. In particular, for any $\hat a_n \in \hat a^\star(G)$ and $a \in a^\star(G)$, we prove
$$\mathbb{E}\left[\int \bigl(C_{n}(\hat a_n,G)-C_{n}(a,G)\bigr)\,d\lambda_n^{(2)}\right]= O\left(\sqrt{\frac{\log n}n}\right),$$

which is equivalently an excess-cost rate of order $O(\sqrt{\log n / n})$. The proof has also been tightened considerably: rather than relying on a loose qualitative plug-in argument, it now directly compares the true and estimated cost functionals and controls the error through the Hellinger-risk bound for transition-density estimation, together with standard norm inequalities and boundedness of the cost function.

Second, we now complement this upper bound with a new minimax lower bound in Theorem 3. There we define the minimax excess-cost risk over contextual MDPs and establish a matching lower bound up to logarithmic factors. Thus, the revised control result is no longer merely an upper bound for a specific plug-in estimator; it is accompanied by a rate-optimality statement showing that, over the model class we study, the statistical order of the bound cannot in general be improved.

More broadly, while our decision rule remains plug-in in nature, this is by design rather than a weakness of the approach. Plug-in methods are well known to enjoy strong optimality properties in statistical decision problems when one first estimates the relevant nuisance object and then optimizes the downstream criterion. A discussion has been added clarifying this.

Our revised Theorems 2 and 3 place the present method in exactly this tradition: the control rule is plug-in, but its excess-risk performance is provably optimal up to logarithmic factors for the contextual MDP class under consideration.

**On Bellman Optimality and Distribution Shift:** In the original version, the connection to offline RL was not developed sufficiently, and that the main technical results were presented primarily through a plug-in control rule based on transition estimation. In response, we substantially revised the manuscript to make the RL content explicit and to clarify the precise scope of our contribution.

Concretely, we added a new section that connects our statistical estimation results to standard offline RL questions. In particular, we now introduce the discounted infinite-horizon value function under a stationary policy $\pi$, together with the associated Bellman equation
$V^\pi = r^\pi + \beta P^\pi V^\pi$,
and its plug-in analogue based on the estimated transition operator $\hat P^\pi$.

We then show that our transition-density estimation bounds imply a quantitative offline policy evaluation guarantee of the form
$$ \mathbb{E}\bigl[\|V^\pi-\hat V^\pi\|\bigr]
\leq
O\left(
\frac{\beta}{(1-\beta)^2}\,\|r^\pi\|_\infty\,
\mathbb{E}\left[H^2(s,\hat s)\right]
\right),
$$
thereby yielding an explicit long-horizon OPE error bound in the discounted contextual MDP setting.

Importantly, and directly in line with the reviewer rmbW's feedback, we also added new results for distribution shift. Proposition 3 establishes a transition-estimation guarantee under shifted data, with additional terms depending on the Hellinger gap $h(s_\star)$ and the discrepancy between invariant measures $\|\nu-\nu^\star\|_{TV}$. Corollary 4 then propagates this shift into offline policy evaluation by bounding the induced value-function error under distribution shift. We agree this was missing from the earlier version, and we have now made it an explicit part of the paper.

At the same time, we revise the paper to be precise about our claims. Our intention is not to suggest that the paper solves the full offline RL problem of finding an optimal policy, but ather, the contribution is to show that nonparametric estimation of contextual transition kernels leads to provable guarantees for downstream control selection and, in the revised version, also for discounted offline policy evaluation, including robustness to distribution shift. We agree that this distinction should have been stated more clearly, and we have revised the exposition accordingly.

---

### Decision · Action_Editor_cpMY · 2026-04-20

**Recommendation:** Accept as is

**Additional Comments:**

This paper is a statistical study of learning contextual MDPs from data. The main idea is extending the framework of T-estimation for transition density estimation, which only requires mild assumptions rather than stationarity and ergodicity. Based on this idea, an oracle inequality in Hellinger distance is given, with theoretical extensions and connections to classical adaptive estimation theory. Finally the paper analyzes a plug-in controller based on this estimation strategy.

The authors and the reviewers had detailed discussions which improved the quality of the paper. All reviewers agreed that the paper is a solid study of a relevant problem, and most concerns from the initial reviews have been sufficiently addressed. Therefore the consensus is to accept this paper. Several remaining weaknesses are noted, including the technical novelty with respect to a related work [1], possible impracticality issue, and the clarity of the proofs. However, in spite of those, the current form of the paper is still a good contribution that well exceeds the acceptance threshold.

To summarize, I would recommend accept as is.

[1] Mathieu Sart, "Estimation of the transition density of a Markov chain", Annales de l’IHP Probabilités et statistiques, vol. 50, pp. 1028–1068, 2014

**Audience:**

Yes

**Audience Explanation:**

The considered problem should be of interest to the subcommunities of TMLR working on nonparametric statistics and sequential decision making.

**Claims And Evidence:**

Yes

**Claims Explanation:**

The paper is a solid contribution to learning the transition density of contextual MDPs. The claims are sufficiently backed up by rigorous analysis, although in the final recommendation certain reviewers suggested improvements on the readability of the proofs.